# The epidermis coordinates thermoresponsive growth through the phyB-PIF4-auxin pathway

Sara Kim[1], Geonhee Hwang[1], Soohwan Kim [1], Thom Nguyen Thi[2], Hanim Kim[3], Jinkil Jeong[4], Jaewook Kim[5], Jungmook Kim[2], Giltsu Choi[3] & Eunkyoo Oh [1✉]

In plants, an elevation in ambient temperature induces adaptive morphological changes including elongated hypocotyls, which is predominantly regulated by a bHLH transcription factor, PIF4. Although *PIF4* is expressed in all aerial tissues including the epidermis, mesophyll, and vascular bundle, its tissue-specific functions in thermomorphogenesis are not known. Here, we show that epidermis-specific expression of *PIF4* induces constitutive long hypocotyls, while vasculature-specific expression of *PIF4* has no effect on hypocotyl growth. RNA-Seq and qRT-PCR analyses reveal that auxin-responsive genes and growth-related genes are highly activated by epidermal, but not by vascular, PIF4. Additionally, inactivation of epidermal PIF4 or auxin signaling, and overexpression of epidermal phyB suppresses thermoresponsive growth, indicating that epidermal PIF4-auxin pathways are essential for the temperature responses. Further, we show that high temperatures increase both epidermal *PIF4* transcription and the epidermal PIF4 DNA-binding ability. Taken together, our study demonstrates that the epidermis regulates thermoresponsive growth through the phyB-PIF4-auxin pathway.

[1] Department of Life Sciences, Korea University, Seoul, Korea. [2] Department of Bioenergy Science and Technology, Chonnam National University, Gwangju 61186, Korea. [3] Department of Biological Sciences, KAIST, Daejeon, Korea. [4] Molecular and Cell Biology Laboratory, Salk Institute for Biological Studies, La Jolla, CA 92037, USA. [5] Faculty of Agriculture, Kyushu University, Fukuoka 819-0395, Japan. ✉email: ekoh@korea.ac.kr

Thermomorphogenesis refers to the morphological changes that occur in plants in response to a rise in ambient temperature[1]. It is characterized by elongated stems and hypocotyls, hyponastic leaf growth, and thinned leaves[1,2]. These morphological changes are considered to enhance leaf cooling capacity by increasing the leaf transpiration rate, thereby helping plants survive under heat stress[3,4]. A bHLH transcription factor, phytochrome interacting factor 4 (PIF4), is a key regulator of thermomorphogenesis[5]. The expression of *PIF4* mRNA is induced when plants are exposed to high temperatures[5]. The resulting PIF4 protein directly activates the expression of the auxin biosynthetic genes (*YUCCA8* (*YUC8*), *TAA1*, and *CYP79B2*), growth-promoting factors (*ATHB2*, *PREs*, and *LNGs*), and brassinosteroid biosynthetic genes that induce hypocotyl growth[6–10]. The high-temperature activation of *PIF4* mRNA expression is mediated by the evening complex (EC)[11,12]. The EC, consisting of ELF3, ELF4, and LUX, binds directly to the promoter of *PIF4* and represses its expression, and it has been reported that high temperatures release this repression by preventing the binding of EC to the *PIF4* promoter[11,13].

In addition to thermomorphogenesis, PIF4 is also involved in a wide range of developmental and environmental responses including photomorphogenesis, shade avoidance responses, hypocotyl gravitropism, phototropism, leaf senescence, and stomatal development[14–20]. PIF4 was first identified as a component of the phytochrome signaling pathway[21]. Light-activated phyB interacts directly with, and phosphorylates, PIF4, resulting in the degradation of the PIF4 protein via the 26S proteasome pathway[20,22]. Recently, it was revealed that phyB functions as a thermosensor for thermomorphogenesis and, therefore, phyB may regulate PIF4 activity in response to changes in ambient temperature as well as light conditions[23,24]. Blue light receptors CRY1 and CYR2 also directly interact with PIF4 in a blue light-dependent manner[25,26]. Interaction with CRY1 represses PIF4 activity by interfering with its transcriptional activation activity[26]. PIF4 also mediates the thermosensory activation of flowering by direct regulation of the expression of *FLOWERING LOCUS T* (*FT*) in a temperature-dependent manner[27], and it suppresses immunity through the negative regulation of defense-related gene expression[28].

Growth regulation by various hormones is dependent on PIF4 activity[15]. In addition to the direct regulation of auxin biosynthetic gene expression, PIF4 also regulates auxin signaling by directly increasing the activities of auxin response factors (ARFs), at least partly by enhancing ARF binding to the promoters of target genes[29,30]. Growth regulation by two other hormones, brassinosteroids (BR) and gibberellic acids (GA), is also mediated by PIF4. BR promotes hypocotyl growth by repressing BIN2 kinase activity and activating the BZR1 transcription factor. BIN2 is a GSK3-like kinase that was reported to directly phosphorylate PIF4 and reduce its stability in the absence of BR signal[31]. The BR-regulated transcription factor, BZR1, directly interacts with PIF4 and this interaction regulates the expression of thousands of genes, promoting hypocotyl growth[8]. It has also been reported that BZR1 activates the expression of *PIF4*[32]. DELLAs, negative regulators of GA signaling pathways, directly interact with both BZR1 and PIF4, and prevent them from binding to target gene promoters[33–35]. GA signals induce the degradation of DELLA repressor proteins, resulting in the de-repression of BZR1 and PIF4 activity, and thereby promoting growth. Consistent with the BR and GA signals being required for PIF4 activity, these phytohormones have been shown to be necessary for thermomorphogenic growth[36].

Although the functional roles of PIF4 in the regulation of growth and development of aerial tissues have been extensively studied, the tissue-specific roles of PIF4 are still poorly understood. It was previously shown that *PIF4* overexpression in

vasculature promotes flowering, and endodermal PIF1, a PIF4 homolog, regulates hypocotyl negative gravitropism[37–39]. Surprisingly, it was reported that epidermal, but not endodermal, phyB activity is required for light regulation of hypocotyl negative gravitropism[40]. Consistent with this fact, epidermal phyB is able to induce the phosphorylation and degradation of endodermal PIFs in response to light, implying that phyB generates an unidentified mobile signal that moves between the epidermis and endodermis[40]. Epidermal phyB was also shown to regulate many light responses, including hypocotyl growth, seed germination, and shade avoidance responses[40]. However, whether epidermal phyB also regulates these responses in a non-cell autonomous manner has not been determined yet.

In this study, we analyzed the tissue-specific roles of PIF4 by generating and characterizing transgenic plants expressing PIF4 or a dominant negative version of PIF4 under the control of either an epidermis- or vasculature-specific promoter. We found that epidermal PIF4, but not vascular PIF4, regulates thermo-responsive hypocotyl growth. It was also observed that the block of epidermal auxin signaling and epidermal overexpression of phyB inhibit thermoresponsive growth. We also found that the expression of *PIF4* mRNA in the epidermis was induced at high temperatures. Our results demonstrate that high temperatures induce thermomorphogenesis through the activation of the PIF4-auxin module in the epidermis.

## Results

**Tissue-specific expression of PIF4-YFP in a *pif* mutant**. PIF4 regulates a wide range of growth and developmental processes such as environmental or hormonal regulation of hypocotyl elongation, hypocotyl phototropism and negative gravitropism, chlorophyll biosynthesis, stomatal development and flowering[5,14,15,19,20,22,41,42]. According to the Cell-Type Specific *Arabidopsis* eFP Browser[43], *PIF4* mRNA is expressed in all aerial tissues, with the highest expression in the vasculature (Supplementary Fig. 1a). To confirm the tissue-specific expression patterns of *PIF4* mRNA, we analyzed β-glucuronidase (GUS) activity in transgenic plants expressing the *GUS* gene controlled by the *PIF4* promoter (PIF4p::GUS). Consistent with the data from the *Arabidopsis* eFP Brower, *PIF4p::GUS* expression was strongly detected in the vasculature of cotyledons (Fig. 1a). A weak GUS signal was also detected in other tissues, including the mesophylls and epidermis, but no GUS signal was detected in the roots (Fig. 1a). To ascertain whether PIF4 in the vasculature is involved in the regulation of various developmental processes including the hypocotyl growth, we generated transgenic plants expressing PIF4-YFP under the control of vascular-specific promoter *SUCROSE TRANSPORTER 2* (*SUC2*) in a *pif1;pif3;pif4;pif5* quadruple mutant (*pifq*) background. We also generated transgenic plants expressing *PIF4-YFP* driven by *MERISTEM LAYER1* (*ML1*) and *CHLOROPHYLL A/B BINDING PROTEIN 3* (*CAB3*) promoter to examine the physiological functions of epidermal and mesophyll PIF4, respectively. These promoters have been successfully used in previous studies to drive the tissue-specific expression of target genes[44–46]. We confirmed that the expression of PIF4-YFP is mainly restricted to the vascular bundles when under the control of the *SUC2* promoter (*SUC2p::PIF4;pifq*) (Fig. 1b). In *ML1p::PIF4;pifq* transgenic plants, PIF4-YFP expression was detected specifically in the epidermis (Fig. 1c). In contrast, the PIF4-YFP signal driven by the *CAB3* promoter was observed in the epidermis in addition to the mesophyll (Supplementary Fig. 1b), probably due to the leaky expression of the *CAB3* promoter. Therefore, we only analyzed *SUC2p::PIF4; pifq* and *ML1p::PIF4;pifq* transgenic plants to determine the tissue-specific functions of PIF4.

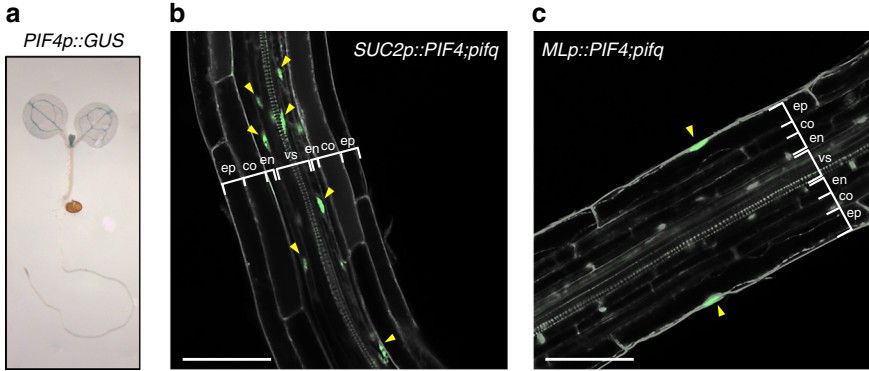

**Fig. 1 Transgenic *Arabidopsis* expressing PIF4-YFP under the epidermis (*ML1*)- or the vasculature (*SUC2*)-specific promoter. a** Histochemical assay of the GUS activity regulated by the *PIF4* promoter. *PIF4p::GUS* transgenic plants were grown under continuous white light for 5 days and harvested for GUS staining. **b, c** Confocal microscopic images showing vascular- or epidermal-specific PIF4-YFP expression (green; yellow arrowheads) in *SUC2p::PIF4;pifq* or *MLp::PIF4;pifq* transgenic plants. PI (gray) was used to counterstain the cell walls. ep, epidermis; co, cortex; en, endodermis; vs, vascular structure. Scale bars = 50 μm.

**Epidermal PIF4 promotes hypocotyl growth**. PIF4 promotes hypocotyl growth in response to various environmental and hormonal signals[15,22]. To determine the tissues in which PIF4 mediates hypocotyl growth, we measured the hypocotyl length of wildtype, *pifq*, *ML1p::PIF4;pifq*, and *SUC2p::PIF4;pifq* seedlings grown under white light. Consistent with previous reports, *pifq* mutants had short hypocotyls compared to those of the wild-type seedlings (Fig. 2a)[41,47]. The hypocotyl lengths of *ML1p::PIF4;pifq* seedlings were substantially longer than those of wild type and *pifq* mutant (Fig. 2a), indicating that epidermal PIF4 is capable of promoting hypocotyl growth. In contrast, vascular expression of *PIF4* driven by the *SUC2* promoter was not able to restore the short hypocotyls of *pifq* mutant seedlings (Fig. 2a), although the level of PIF4 in *SUC2p::PIF4;pifq* plants was higher than those in wild type and *ML1p::PIF4;pifq* plants (Fig. 2b).

A recent study showed that the circadian clock activity in the epidermal cells is important in the regulation of hypocotyl and petiole elongation[48]. We confirmed that while perturbation of the epidermal clock system by overexpression of *CCA1* specifically in the epidermis (*CER6p::CCA1*) promotes the hypocotyl growth, neither vascular (*SUC2p::CCA1*) nor mesophyll (*CAB3p::CCA1*) clock perturbation had a significant effect on hypocotyl growth[48] (Fig. 2c). A previous study showed that *PIF4* expression is constitutively high in CCA1-overexpressing plants[49]. Intriguingly, *PIF4* expression was highly elevated in the *CER6p::CCA1*, but showed only a marginal increase in *SUC2p::CCA1* seedlings compared to those in wild-type seedlings (Fig. 2d), suggesting the hypocotyl elongation seen in *CER6p::CCA1* plants might be attributable to the increased *PIF4* levels in the epidermis. In support of this, the down-regulation of *PIF4* expression by artificial microRNA against *PIF4* (*PIF4mi*) suppressed the long hypocotyls of *CER6p::CCA1* (Supplementary Fig. 2). Therefore, these results provide further support that epidermal PIF4 promotes hypocotyl growth.

To obtain clues for molecular mechanisms underlying the epidermal PIF4 regulation of hypocotyl growth, we performed RNA-Seq analysis of wild type, *pifq*, *ML1p::PIF4;pifq*, and *SUC2p::PIF4;pifq* seedlings grown under white light for 5 days. The RNA-Seq analysis identified 1281 differentially expressed genes (DEGs) between wild type and *pifq* seedlings (Supplementary Data 1). We refer to these DEGs as the global PIF quartet (PIF1, PIF3, PIF4, and PIF5)-regulated genes. We also compared the transcriptome of *pifq* with either *ML1p::PIF4;pifq* or *SUC2p::PIF4;pifq* and identified 575 epidermal PIF4 (ML1:PIF4)-regulated genes and 1765 vascular PIF4 (SUC2:PIF4)-regulated genes (Supplementary Data 1).

Thereafter, using Gene Set Enrichment Analysis (GSEA), we examined how the global PIF quartet-regulated genes are regulated by each tissue-specific PIF4 (Fig. 2e, f, Supplementary Fig. 3). The global PIF quartet-activated gene set shows a more significant positive enrichment in *ML1p::PIF4;pifq* than that in *SUC2p::PIF4; pifq* when compared to *pifq* (Supplementary Fig. 3a, b), suggesting that the PIF quartet-activated genes are more robustly activated by epidermal PIF4 than by vascular PIF4. In contrast, the global PIF quartet-repressed gene set shows a more significant negative enrichment in *SUC2p::PIF4;pifq* than that in *ML1p::PIF4;pifq* when compared to *pifq* (Supplementary Fig. 3c, d). This suggests that vascular PIF4 may play a dominant role in the regulation of PIF quartet-repressed genes. On the other hand, we tested the statistical significance of overlaps between each of the gene sets shown in Fig. 2e, f (Supplementary Data 1). The global PIF quartet-regulated genes showed a highly significant overlap with either of the epidermal PIF4- or vascular PIF4-regulated genes (Supplementary Data 1). However, the overlap between epidermal PIF4-regulated genes and vascular PIF-regulated genes were far less significant in both up- and down-regulated DEGs, showing an overlap enriched only 2.5-fold than that by random chance in both cases (Supplementary Data 1). The distinct DEGs of epidermal PIF4 and vascular PIF4 indicate high tissue-dependency in PIF4 target gene regulation, and also validate robust tissue-specific expression of PIF4 by *ML1* and *SUC2* promoters (Fig. 1b, c).

Gene ontology (GO) analyses revealed that GO terms including "response to auxin stimulus," "cell wall organization," and "cell growth" were highly enriched only in epidermal PIF4-activated genes (Fig. 2g). The results are very consistent with the positive role of epidermal PIF4 in hypocotyl growth (Fig. 2a), suggesting that epidermal PIF4 regulates hypocotyl growth through the auxin pathway. Interestingly, "iron ion transporter" and "response to nitrate" GO terms were highly enriched in the vascular PIF4-repressed genes (Fig. 2g). Thus, it is possible that the vascular PIF4 may mediate these iron and nitrogen responses. It was previously shown that PIF4 negatively regulates defense responses[28]. In GO analyses, the GO term of defense responses was enriched in both epidermal PIF4-repressed and vascular PIF4-repressed genes (Fig. 2g). Thus, it appears that both epidermal and vascular PIF4 participate in the regulation of defense responses. To confirm that epidermal PIF4 modulates auxin responses, we compared epidermal or vascular PIF4-regulated genes with the auxin-regulated genes in hypocotyls[50]. Of 266 epidermal PIF4-activated genes, 75 (6.1 expected randomly) were auxin-activated genes (Fig. 2h, Supplementary

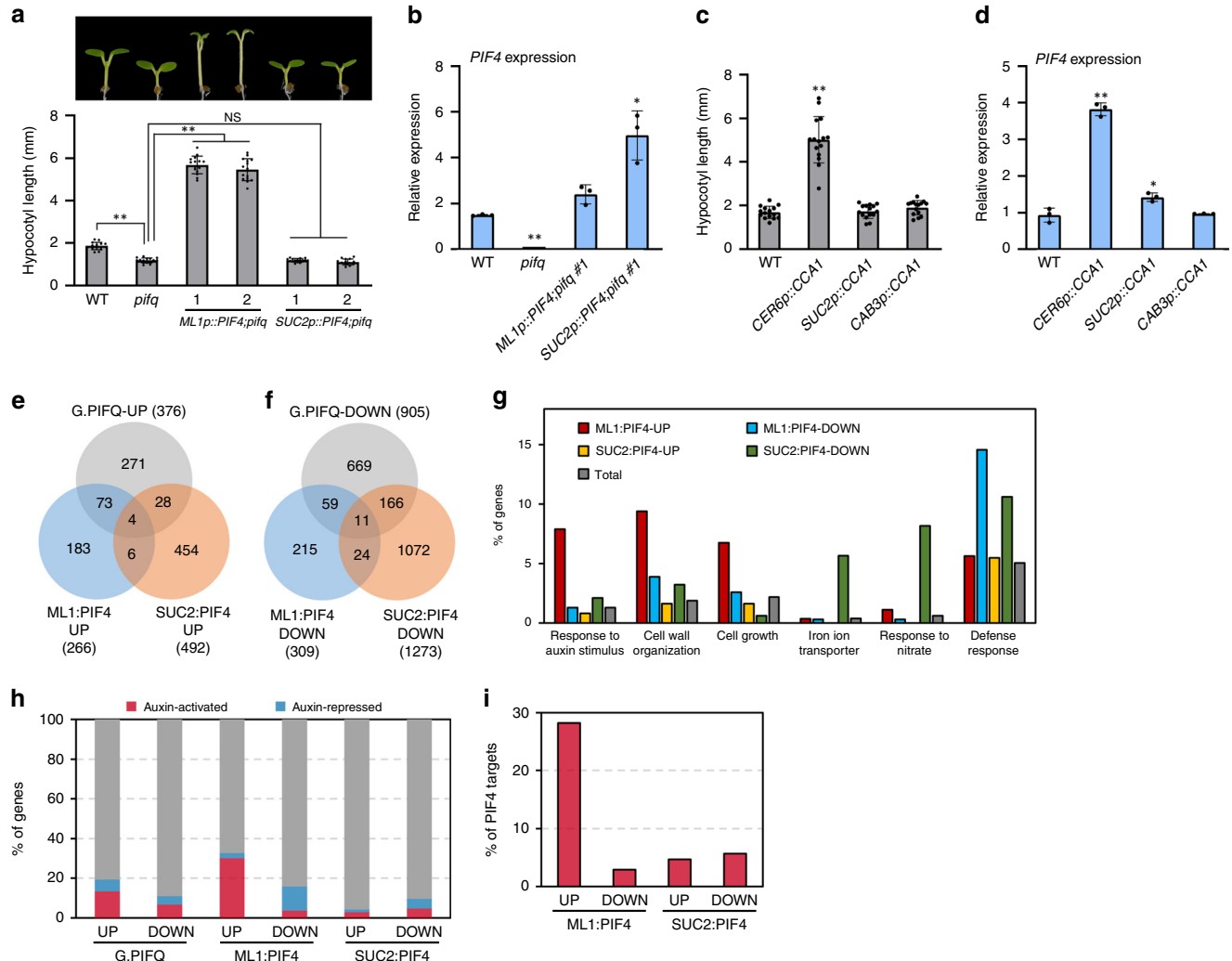

**Fig. 2 Epidermal PIF4 promotes hypocotyl growth. a** Hypocotyl lengths of seedlings grown under the continuous white light at 20 °C for 7 days. Error bars indicate s.d. ($n = 15$ plants). **$P<0.01$ (two-tailed Student's $t$-test); NS, not significant (two-tailed Student's $t$-test $P \geq 0.05$). **b** qRT-PCR analysis of PIF4. Seedlings were grown in the same conditions as **a**. The expression levels of PIF4 were normalized to those of PP2A. Error bars indicate s.d. ($n = 3$). *$P < 0.05$ and **$P < 0.01$ (two-tailed Student's $t$-test). **c** Hypocotyl lengths of seedlings grown under continuous white light at 20 °C for 7 days. Error bars indicate s.d. ($n = 15$ plants). **$P < 0.01$ (two-tailed Student's $t$-test). **d** PIF4 expression levels in the seedlings grown in the same growth conditions as **c**. The expression levels of PIF4 were normalized to those of PP2A. Error bars indicate s.d. ($n = 3$). *$P < 0.05$ and **$P < 0.01$ (two-tailed Student's $t$-test). **e** The Venn diagram shows the overlap between genes that are up-regulated by global PIF quartet (G.PIFQ), ML1 promoter-driven PIF4 (ML1:PIF4 UP), and SUC2 promoter-driven PIF4 (SUC2:PIF4 UP). **f** The Venn diagram shows the overlap between genes that are down-regulated by the global PIF quartet, ML1:PIF4, and SUC2:PIF4. **g** GO analyses of ML1:PIF4-regulated and SUC2:PIF4-regulated genes. Total: Arabidopsis total genes. **h** Percentage of auxin (picloram)-activated or repressed genes in global PIF quartet-regulated or ML1:PIF4-regulated or SUC2:PIF4-regulated genes. **i** Percentage of PIF4 direct target genes in ML1:PIF4-regulated or SUC2:PIF4-regulated genes. Source data are provided as a Source Data file.

Data 2). In contrast, only 14 of 492 vascular PIF4-activated genes (11.2 expected randomly) were auxin-activated genes (Fig. 2h). In addition, auxin-repressed genes were enriched in epidermal PIF4-repressed genes (38 of 309, 4.3 expected randomly) (Fig. 2h, Supplementary Data 2). These results provide further support that epidermal PIF4 activates auxin responses.

We also compared epidermal or vascular PIF4-regulated genes with previously identified PIF4 direct target genes[51]. A large portion (75 of 266) of epidermal PIF4-activated genes (11.2 expected randomly), but not epidermal PIF4-repressed and vascular PIF4-activated/repressed genes, were PIF4 direct target genes (Fig. 2i, Supplementary Data 3). Of 75 epidermal PIF4-regulated PIF4 direct target genes, 35 were auxin-activated genes (Supplementary Data 3), suggesting that epidermal PIF4 directly regulates auxin responses.

**Epidermal PIF4 regulates thermoresponsive hypocotyl growth.** Next, we examined whether epidermal or vascular PIF4 activity regulates thermoresponsive hypocotyl growth. The hypocotyl lengths of wild-type plants exposed to 28 °C for 3 days were approximately five times longer than those grown constitutively at 20 °C (Fig. 3a). In contrast, the hypocotyl growth in the pifq mutant was almost insensitive to the increase in ambient temperature (Fig. 3a). The hypocotyls of ML1p::PIF4;pifq plants grown at 20 °C were longer than those of wild-type plants also grown at 20 °C, and the lengths were further increased by exposure to high temperatures (Fig. 3a). In contrast, the hypocotyl growth of SUC2p::PIF4;pifq plants was only marginally increased by high temperature (Fig. 3a), like that of pifq. These results show that epidermal PIF4 activity is essential for the hypocotyl growth response to high temperatures.

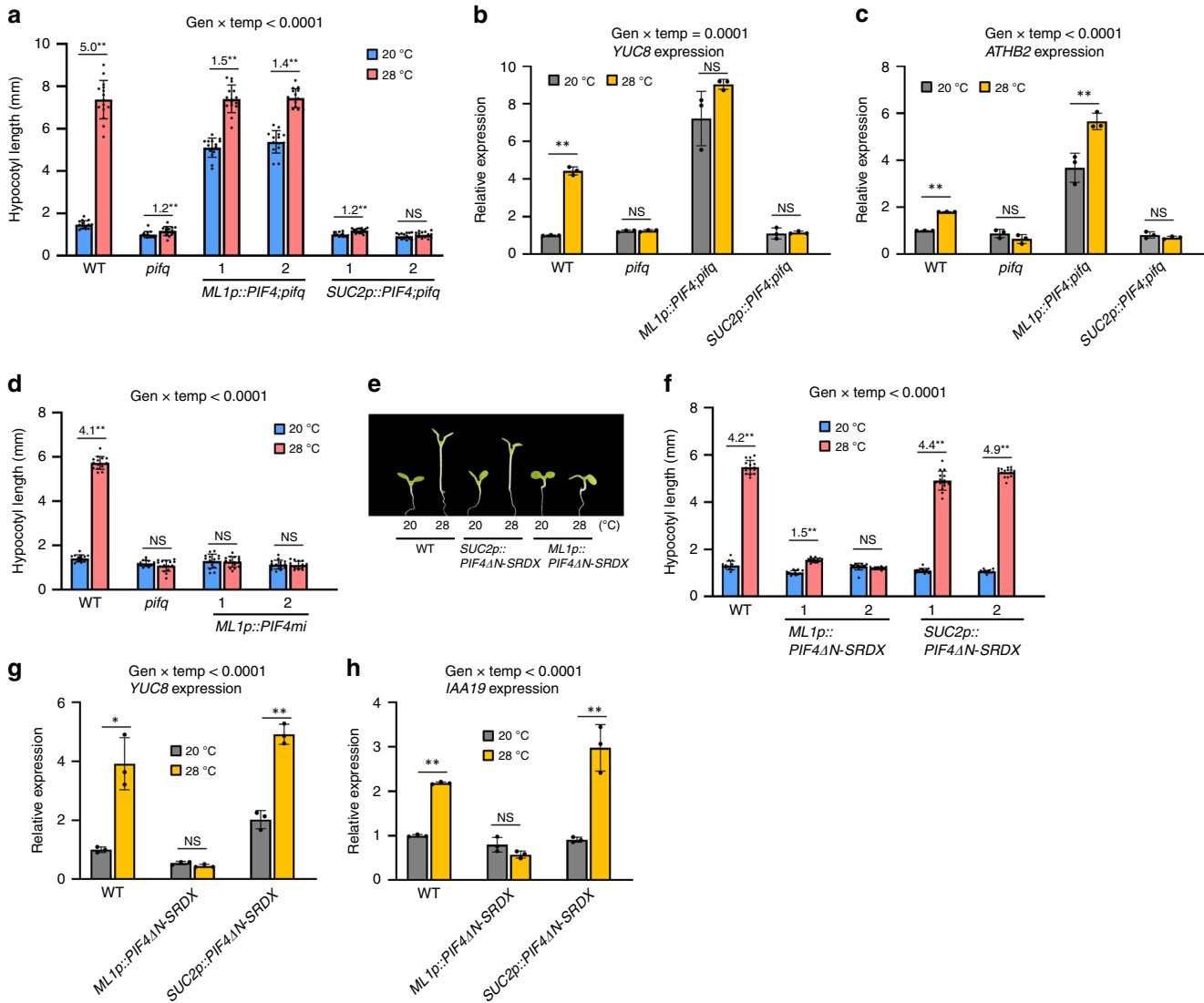

**Fig. 3 Epidermal PIF4 activity is required for thermoresponsive hypocotyl growth. a** Hypocotyl lengths of seedlings grown under continuous white light at 20 °C for 7 days or 20 °C for 4 days followed by 28 °C for 3 days. Error bars indicate s.d. (*n* = 15 plants). The *P*-value for the interaction term (genotype x temperature) calculated by two-way ANOVA is shown at the top. Black asterisks above the bars (in a-h) indicate significant differences (**$P$ < 0.01 and *$P$ < 0.05, two-tailed Student's *t*-test). NS, not significant ($P \geq 0.05$). Numbers above the bars (in a, d, f) indicate the ratio of hypocotyl lengths of seedlings grown at two different temperatures (28 °C/20 °C). **b, c** qRT-PCR analyses of *YUC8* and *ATHB2* expression. Seedlings were grown in 12 h light/12 h dark cycles (12 L:12D) at 20 °C for 4 days and transferred under the continuous white light on 5th day. The growth temperature was increased to 28 °C or kept at 20 °C for 4 h at Zeitgeber Time (ZT) 20-24 before harvesting for total RNA extraction. Gene expression levels were normalized to *APX3* and presented as values relative to that of the wild type (WT) kept at 20 °C. Error bars indicate s.d. (*n* = 3). The *P*-value for the interaction (genotype x temperature) is shown at the top. **d** Hypocotyl lengths of WT, *pifq*, and *ML1p::PIF4mi* seedlings grown under continuous white light at 20 °C for 7 days or 20 °C for 4 days followed by 28 °C for 3 days. Error bars indicate s.d. (*n* = 15 plants). The *P*-value for the interaction (genotype x temperature) is shown at the top. **e, f** Hypocotyl lengths of WT, *ML1p::PIF4ΔN-SRDX*, and *SUC2p::PIF4ΔN-SRDX* seedlings grown under continuous white light at 20 °C for 7 days or 20 °C for 4 days followed by 28 °C for 3 days. Representative seedlings are shown in **e** and error bars in **f** indicate s.d. (*n* = 15 plants). The *P*-value for the interaction (genotype x temperature) is shown at the top. **g, h** qRT-PCR analyses of *YUC8* and *IAA19*. WT and transgenic seedlings were grown in 12 h light/12 h dark cycles (12 L:12D) at 20 °C for 4 days and transferred under the continuous white light on 5th day. The growth temperature was increased to 28 °C or kept at 20 °C for 4 h at ZT20-24 before harvesting for total RNA extraction. Gene expression levels were normalized to *APX3* and presented as values relative to that of wild type (WT) kept at 20 °C. Error bars indicate s.d. (*n* = 3). The *P*-values for the interaction (genotype x temperature) are shown at the top. Source data are provided as a Source Data file.

PIF4 has previously been shown to directly activate auxin biosynthesis and signaling to induce hypocotyl growth at high temperatures[6–8]. Consistent with this fact, RNA-Seq analyses found that epidermal PIF4-regulated genes include many auxin-responsive genes in hypocotyls (Fig. 2h). To further examine whether epidermal PIF4 promotes hypocotyl growth through the auxin pathway, we measured the expression levels of the auxin

biosynthetic gene *YUCCA8* (*YUC8*) and auxin-responsive genes (*IAA19* and *ATHB2*) in seedlings that were grown continuously at 20 °C or were shifted to 28 °C for 4 h. These genes are also PIF4 direct target genes[8,51]. Consistent with the previous studies[7,8], the expression of *YUC8*, *IAA19*, and *ATHB2* was highly induced by exposure to high temperatures in wild-type plants, but not in *pifq* mutant plants (Fig. 3b, c, Supplementary Fig. 4). The expression

levels of these same genes in *ML1p::PIF4;pifq* plants grown at 20 °C showed a marked increase, to similar levels observed in the wild-type plants exposed to 28 °C (Fig. 3b, c, Supplementary Fig. 4), indicating that epidermal PIF4 activates auxin responses, even at 20 °C. While *ATHB2* expression in *ML1p::PIF4;pifq* plants was further increased by exposure to high temperature, the expression of *YUC8* and *IAA19* was not significantly affected (Fig. 3b, c, Supplementary Fig. 4). The additional increase in *ATHB2* expression by high temperature may account for the longer hypocotyls of *ML1p::PIF4;pifq* seedlings grown at 28 °C compared to those grown at 20 °C (Fig. 3a). In contrast to *ML1p::PIF4;pifq* plants, *YUC8*, *IAA19*, and *ATHB2* expression was not induced by exposure to high temperature in *SUC2p::PIF4;pifq* plants, similar to the *pifq* mutant (Fig. 3b, c, Supplementary Fig. 4) and consistent with their thermo-insensitive hypocotyl growth phenotype (Fig. 3a). Together, these results show that epidermal, but not the vascular, PIF4 activates auxin responses, thereby triggering thermomorphogenesis.

To further confirm that thermomorphogenesis is dependent on epidermal PIF4 activity, we generated transgenic plants expressing an epidermis-specific *PIF4* artificial microRNA (*ML1p::PIF4mi*) and examined thermoresponsive hypocotyl growth in the transgenic plants. We found that *PIF4* mRNA expression was reduced by approximately half compared to that in wild-type plants (Supplementary Fig. 5), likely due to the *PIF4mi*- induced reduction in epidermal *PIF4* mRNA levels. The hypocotyl growth of two independent *ML1p::PIF4mi* plants were insensitive to the high-temperature exposure (Fig. 3d), similar to the *pifq* mutant, supporting that epidermal PIF4 activity is required for high temperature-induced hypocotyl growth. However, it is possible that the small microRNAs may have moved to neighboring cells from the cells where the microRNAs were initially biosynthesized;[52] therefore, the thermo-insensitive hypocotyl growth seen in *ML1p::PIF4mi* plants could be attributable to reduced PIF4 levels in neighboring tissues other than the epidermis.

To exclude this possibility, we utilized PIF4, with the N-terminal deleted, fused to the EAR repression domain (PIF4ΔN-SRDX) to inhibit PIF4 activity in a tissue-specific manner[53]. PIF4ΔN fused to the strong transcriptional repressor SRDX is expected to inhibit endogenous PIF4 activity by competitively binding to the promoters of PIF4 target genes and repressing their expression. Deletion of the PIF4 N-terminus, that includes the active phytochrome binding motif (APB), was shown to increase the PIF4 stability under light[20], enhancing the negative effects of PIF4-SRDX on endogenous PIF4 activity. The PIF4ΔN-SRDX chimera driven by the *UBQ10* promoter (*UBQ10p::PIF4ΔN-SRDX*), which is strongly active in all aerial tissues, greatly suppressed hypocotyl growth at high temperatures (Supplementary Fig. 6a, b), indicating that the PIF4ΔN-SRDX chimera represses endogenous PIF4 activity, as expected. To repress PIF4 activity only in the epidermis and vasculature, we expressed the PIF4ΔN-SRDX protein under the control of the *ML1* and *SUC2* promoters, respectively (*ML1p::PIF4ΔN-SRDX* and *SUC2p::PIF4ΔN-SRDX*). In agreement with the results obtained from the analyses of *ML1p::PIF4mi*, hypocotyl growth in *ML1p::PIF4ΔN-SRDX* transgenic plants was not significantly enhanced by high temperatures (Fig. 3e, f). In contrast, *SUC2* promoter-driven *PIF4ΔN-SRDX* had no significant effect on high temperature-induced hypocotyl growth (Fig. 3e, f). Consistent with thermo-insensitive hypocotyl growth, the expression of the PIF4 target genes *YUC8* and *IAA19* was not induced by exposure to high temperature in *ML1p::PIF4ΔN-SRDX* transgenic plants, in contrast to that observed in wild-type and *SUC2p::PIF4ΔN-SRDX* plants (Fig. 3g, h). Together, these results demonstrate that epidermal, but not vascular, PIF4 regulates thermoresponsive hypocotyl growth.

**Epidermal auxin regulates thermoresponsive hypocotyl growth**. We next tested whether epidermal PIF4-activated auxin responses are essential for high temperature-induced hypocotyl growth. We generated transgenic plants expressing a stabilized mutant form of IAA3/SHY2 (shy2–2) under the control of the *ML1* promoter, in which the epidermal expression of shy2–2 blocks the transcriptional responses to auxin in the epidermis[54], and analyzed thermoresponsive hypocotyl growth. As shown in Fig. 4a, blocking auxin responses in the epidermis significantly reduced the hypocotyl growth response to an elevation in ambient temperature, indicating that epidermal auxin responses are required for high temperature-induced hypocotyl elongation. Consistent with our observation, a previous study showed that epidermal expression of a stabilized mutant version of IAA17 protein suppressed the hypocotyl growth response to both shade and high temperatures[55]. These results suggest that the environmental regulation of hypocotyl growth may be determined by epidermal auxin responses.

**Epidermal phyB inhibits temperature-induced hypocotyl growth**. The light-activated phyB represses PIF4 activity through both the proteasome-mediated degradation of PIF4 and the prevention of PIF4 from binding to target genes[20,56]. It was previously shown that epidermal phyB represses hypocotyl growth[40]. Given our results indicating that epidermal PIF4 promotes hypocotyl growth, it is highly likely that epidermal phyB impacts on hypocotyl growth by repressing the epidermal PIF4 activity. If this is the case, the overexpression of epidermal phyB suppresses thermomorphogenesis. To test this hypothesis, we generated transgenic plants expressing *PHYB* under control of the *ML1* promoter in *phyB* mutant background (*phyB-9*). Total *PHYB* levels were higher in *ML1p::PHYB;phyB-9* than those in the wild type (Supplementary Fig. 7), implicating that epidermal phyB is overexpressed in *ML1p::PHYB;phyB-9*. As expected, while the hypocotyl lengths in wild-type seedlings were increased more than two times by high temperature, those in *ML1p::PHYB;phyB-9* transgenic plants were only marginally increased (Fig. 4b). Consistent with the defect in thermoresponsive hypocotyl growth, the expression of PIF4 target gene (*YUC8*) was largely repressed and insensitive to exposure to high temperature in *ML1p::PHYB;phyB-9* transgenic plants (Fig. 4c). In contrast, the expression of *PIF4* was similarly induced by high-temperature exposure in wild type and *ML1p::PHYB;phyB-9* transgenic plants (Fig. 4d). Taken together, these results suggest that the overexpressed epidermal phyB suppresses thermomorphogenesis through the inactivation of epidermal PIF4 post-transcriptionally, probably in a cell-autonomous manner.

**Role of epidermal and vascular PIF4 in skotomorphogenesis**. In the dark, PIF4, together with three additional PIFs (PIF1, PIF3, and PIF5), promotes skotomorphogenesis including hypocotyl growth, a closed cotyledon, and hypocotyl negative gravitropism[47,57]. To examine whether epidermal and vascular PIF4 are also involved in these developmental processes, we examined hypocotyl length, cotyledon opening, and hypocotyl negative gravitropism in *ML1p::PIF4;pifq* and *SUC2p::PIF4;pifq* transgenic plants in the dark. Surprisingly, and in contrast to the light conditions, the hypocotyl lengths of both *ML1p::PIF4;pifq* and *SUC2p::PIF4;pifq* plants were shorter than those of wild-type plants, similar to the *pifq* mutant (Fig. 5a). This indicates that epidermal or vascular expression of *PIF4* is not sufficient to promote hypocotyl growth in etiolated seedlings. This also indicates that the mechanisms underlying PIF-mediated hypocotyl elongation are different under light and dark conditions. While both wild type and *ML1p::PIF4;pifq* plants had closed cotyledons,

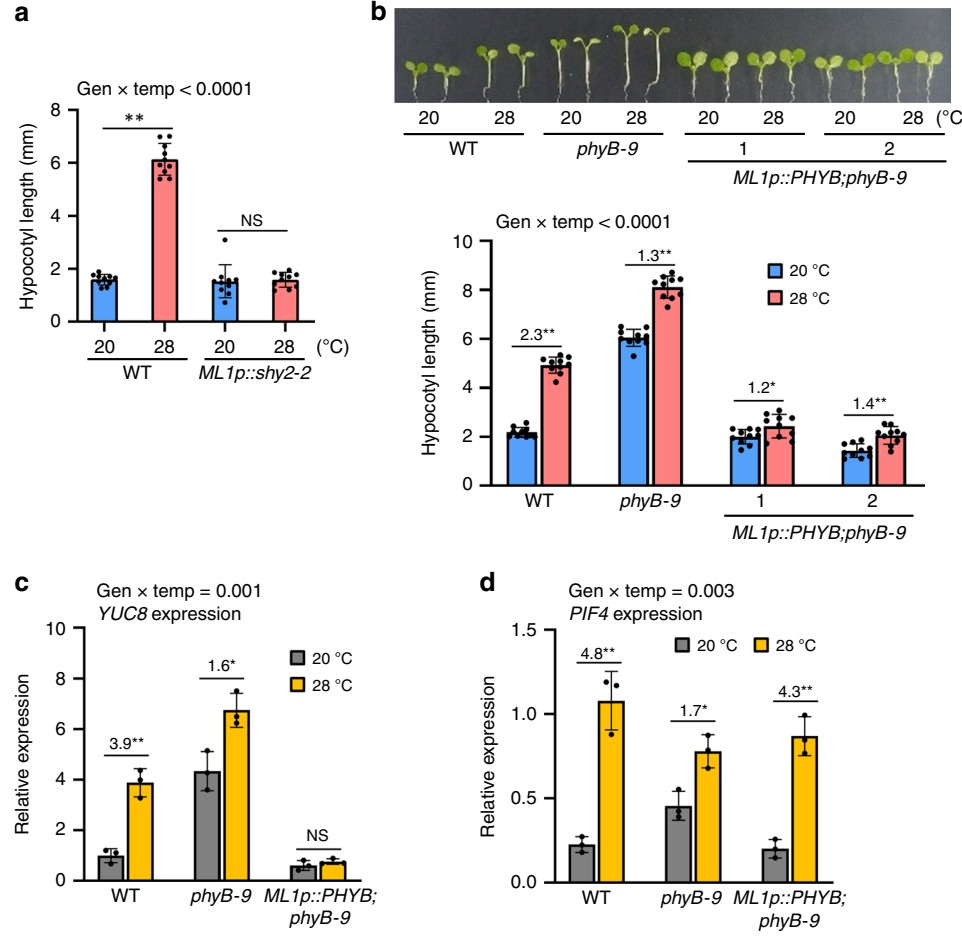

**Fig. 4 Epidermal auxin and phyB regulate high temperature-induced hypocotyl growth. a** Hypocotyl lengths of the wild-type (WT) and *ML1p::shy2-2* seedlings grown in the same conditions described in Fig. 3a. Error bars indicate s.d. (*n* = 10 plants). The *P*-value for the interaction term (genotype x temperature) calculated by two-way ANOVA is shown at the top. Black asterisks above the bars (in a-d) indicate significant differences (**\**P* < 0.01 and *\**P* < 0.05, two-tailed Student's *t*-test). NS, not significant (*P* ≥ 0.05). **b** Hypocotyl lengths of WT, *phyB-9*, and *ML1p::PHYB;phyB-9* seedlings grown in the same conditions described in **a**. Error bars indicate s.d. (*n* = 10 plants). Representative seedlings are shown in the upper panel. The numbers above the bars indicate the ratio of hypocotyl lengths of seedlings grown at two different temperatures (28 °C/20 °C). The *P*-value for the interaction (genotype x temperature) is shown at the top. **c**, **d** qRT-PCR analyses of *YUC8* and *PIF4*. Seedlings were grown in 12 h light/dark cycles (12 L:12D) at 20 °C for 4 days and transferred to the continuous white light on the 5th day. The growth temperature was increased to 28 °C or kept at 20 °C for 4 h at ZT20-24 before harvesting for total RNA extraction. Gene expression levels were normalized to *APX3* and presented as values relative to that of the WT kept at 20 °C. Error bars indicate s.d. (*n* = 3). The *P*-value for the interaction (genotype x temperature) is shown at the top. The numbers above the bars indicate the ratio of gene expression levels at two different temperatures (28 °C/20 °C). Source data are provided as a Source Data file.

*SUC2p::PIF4;pifq* and *pifq* plants had opened cotyledons (Fig. 5b), showing that epidermal PIF4 is involved in inhibiting the cotyledon opening of etiolated seedlings. Similarly to hypocotyl elongation, both epidermal and vascular PIF4 failed to rescue the hypocotyl negative gravitropism of *pifq* mutants (Fig. 5c), consistent with previous studies showing that endodermal PIFs mediate negative hypocotyl gravitropism[37,57].

**Vascular PIF4 plays a dominant role in flowering.** PIF4 is also involved in the thermosensory flowering pathway by directly activating the expression of *FLOWERING LOCUS T* (*FT*) at high temperatures. It has previously been reported that PIF4 in the vasculature is capable of promoting flowering[38,39]. To ascertain whether epidermal PIF4 is also able to induce flowering, we measured the flowering time in *ML1p::PIF4;pifq* and *SUC2p::PIF4;pifq* plants under a non-inductive photoperiod (8-h light: 16-h dark) and high temperature (27 °C) condition, where PIF4 is known to promote flowering[27]. Consistent with the previous reports[38,39], *SUC2* promoter-driven *PIF4*

expression resulted in very early flowering, whereas epidermal *PIF4* expression only slightly accelerated flowering compared to that in wild-type plants (Supplementary Fig. 8a), indicating that vascular PIF4 plays a dominant role in the promotion of flowering. To further verify this result, we determined the flowering time in *ML1p::PIF4ΔN-SRDX* and *SUC2p::PIF4ΔN-SRDX* transgenic plants under an inductive photoperiod (16 h-light: 8 h-dark). As shown in Supplementary Fig. 8b, while inactivation of PIF4 specifically in the vascular tissues (*SUC2p::PIF4ΔN-SRDX*) greatly delayed flowering, inactivation of PIF4 in the epidermis (*ML1p::PIF4ΔN-SRDX*) did not significantly affect flowering time. Consistent with the late-flowering phenotypes, *FT* expression levels showed a dramatic reduction in *SUC2p::PIF4ΔN-SRDX* plants (Supplementary Fig. 8c). Although the flowering time was not significantly affected by the epidermal inactivation of PIF4 under long days, the *FT* expression levels in *ML1p::PIF4ΔN-SRDX* plants were approximately half that seen in wild-type plants (Supplementary Fig. 8c). This indicates that epidermal PIF4 may also affect flowering time under certain

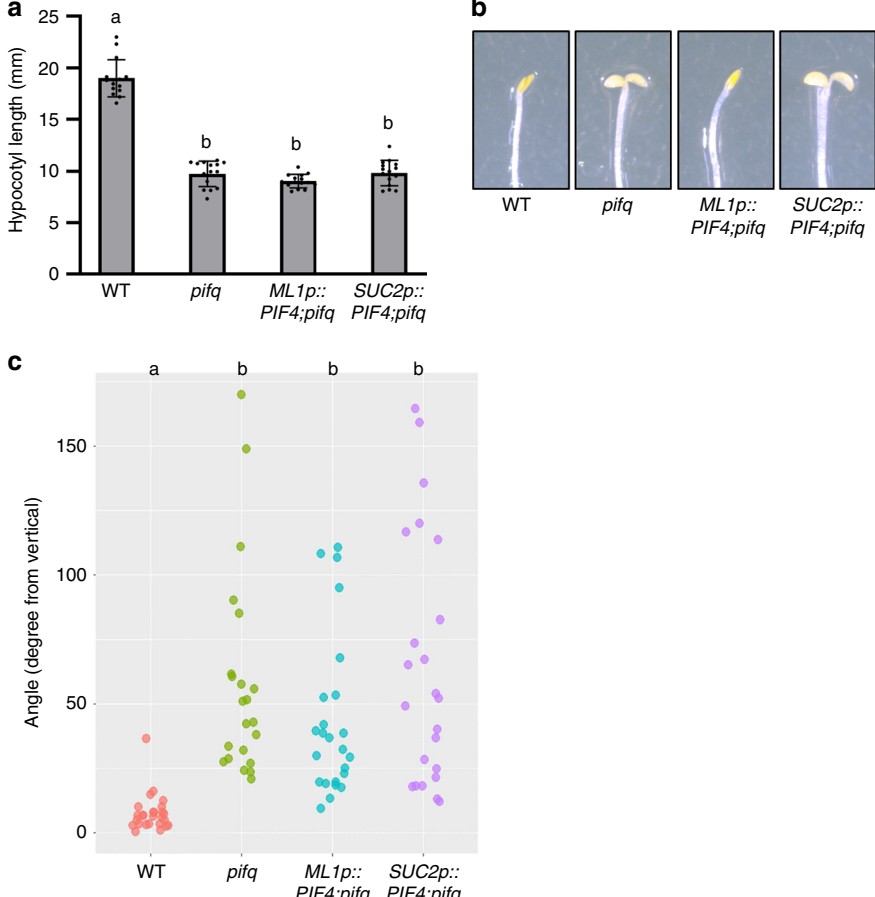

**Fig. 5 Neither epidermal nor vascular PIF4 regulates hypocotyl elongation and hypocotyl negative gravitropism in the dark. a** Hypocotyl lengths of WT, *pifq*, *ML1::PIF4;pifq*, and *SUC2p::PIF4;pifq* seedlings grown in the dark for 5 days. Error bars indicate s.d. ($n = 15$ plants). Letters above the bars indicate significant differences based on one-way ANOVA and Tukey's test ($P < 0.05$). **b** Epidermal PIF4, but not vascular PIF4, represses cotyledon opening in the dark. Representative seedlings grown in the dark for 5 days are shown. **c** Scatter plot of the measurement of hypocotyl negative gravitropism in the WT, *pifq*, *ML1p::PIF4;pifq*, and *SUC2p::PIF4;pifq* seedlings. Seedlings were grown in the dark on vertically oriented agar surfaces and angles between the growing direction of hypocotyls and the opposing direction of gravity were measured. Letters indicate significant differences based on one-way ANOVA and Tukey's test ($P < 0.05$). Source data are provided as a Source Data file.

conditions, although to a much lesser extent than that by vascular PIF4. Taken together, these results show that vascular PIF4 has a greater effect on the regulation of flowering than that by epidermal PIF4.

**Epidermal PIF4 DNA-binding is enhanced by high temperatures**. The stability of the PIF4 protein is known to be regulated by light. Upon irradiation, light-activated phytochromes interact directly with PIF4, and the interaction induces PIF4 phosphorylation[20]. The phosphorylated form of PIF4 is then ubiquitinated and degraded via the 26 S proteasome pathway[20]. To examine whether the stability of PIF4 in the epidermis is degraded by light-activated phytochrome signals, we determined PIF4 protein levels in *ML1p::PIF4;pifq* transgenic plants grown in the dark for 5 days or irradiated by red light for 2 h after the 5-day incubation in the dark. Figure 6a shows that the levels of PIF4-YFP were significantly reduced after 2 h of red-light irradiation, indicating that light induces the degradation of PIF4 in the epidermis. Consistent with this result, the levels of PIF4-YFP were increased when the light-grown *ML1p::PIF4;pifq* plants were transferred to darkness (Supplementary Fig. 9). It was recently reported that epidermal phyB is sufficient for many light-mediated responses, including photomorphogenesis[40]. Thus, it is highly likely that

epidermal phyB mediates light-induced PIF4 degradation in a cell-autonomous manner.

Our genetic analyses showed that epidermal PIF4 controls thermoresponsive hypocotyl growth, suggesting that the epidermal PIF4 level or activity increases in response to a rise in ambient temperature to promote thermomorphogenesis. To test whether this is indeed the case, we first determined whether the level of epidermal PIF4 driven by the *ML1* promoter (ML1p::PIF4) is affected by ambient temperature changes. The PIF4-YFP protein levels in *ML1p::PIF4-YFP;pifq* plants were not significantly altered either by 6 h or 24 h of high-temperature exposure under light (Fig. 6b). Similarly, the dark stabilization of epidermal PIF4 was not accelerated at high temperatures (Supplementary Fig. 9). These results suggest that epidermal PIF4 protein stability may not be affected by high temperatures.

Next, we determined whether the DNA-binding of epidermal PIF4 is altered by changes in ambient temperatures. A temperature shift from 16 °C to 28 °C was used because the difference in *YUC8* expression was significant between these two temperature conditions (Supplementary Fig. 10). Chromatin immunoprecipitation (ChIP) assays confirmed that epidermal PIF4 directly binds to the promoters of *YUC8*, *IAA19*, and *ATHB2* (Fig. 6c), which is consistent with the activation of these genes by epidermal PIF4 (Fig. 3b, c). The binding of epidermal

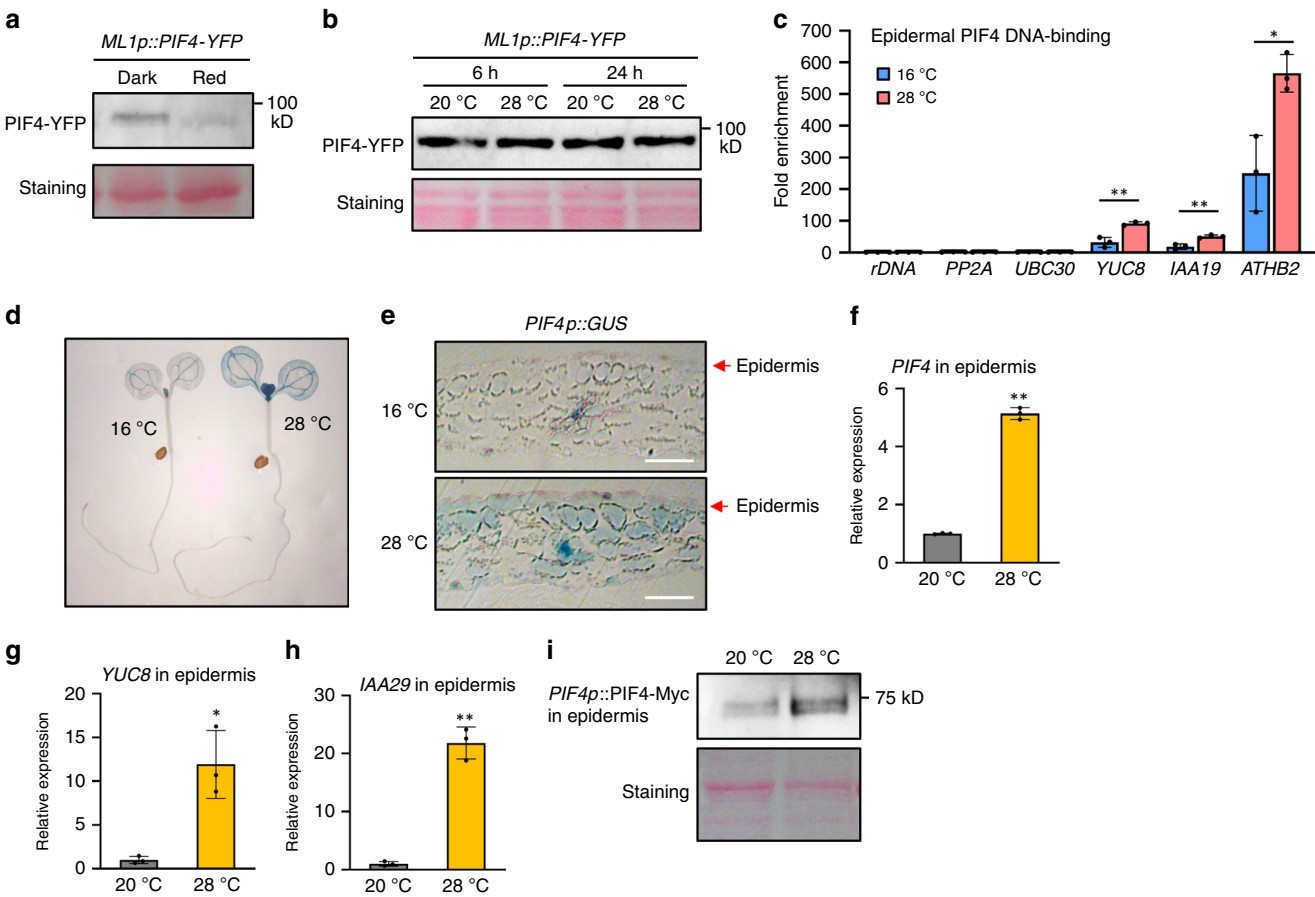

**Fig. 6 Both epidermal PIF4 DNA-binding and epidermal *PIF4* mRNA expression is increased at high temperatures. a** Red light induces the degradation of epidermal PIF4 protein. *ML1p::PIF4-YFP;pifq* seedlings were grown in the dark and transferred under red light for 2 h before harvesting. Immunoblotting was probed using an anti-GFP antibody. Ponceau S staining is shown for equal loading. **b** High temperature effects on the stability of epidermal PIF4 protein. *ML1p::PIF4-YFP;pifq* seedlings were grown under white light constitutively at 20 °C or grown at 20 °C and shifted to 28 °C for 6 or 24 h. Immunoblotting was probed using an anti-GFP antibody. Ponceau S staining is shown for equal loading. **c** ChIP assays showing epidermal PIF4 DNA-binding is increased by high temperatures. WT and *ML1p::PIF4;pifq* seedlings kept at 16 °C or shifted to 28 °C for 24 h were used for ChIP assays. Enrichment of DNA was calculated as the ratio between *ML1p::PIF4;pifq* and WT control, normalized to that of the *rDNA* gene as an internal reference. Error bars indicate s.d. (n = 3). ** P < 0.01 and * P < 0.05 (two-tailed Student's t-test). **d, e** The effects of high temperatures on the expression of PIF4p::GUS. *PIF4p::GUS* seedlings were grown under white light constitutively at 16 °C or grown at 16 °C and shifted to 28 °C for 24 h. Cross-sections of cotyledons of seedlings are shown in **e**. Scale bars = 50 μm. **f–h** The expression of *PIF4* and its target genes is increased by high temperatures in the epidermis. Wild-type seedlings were grown 20 °C or were shifted to 28 °C for 24 h. Total RNA was extracted from the isolated epidermal tissues. Gene expression levels were normalized to *APX3* and presented as values relative to that of wild type at 20 °C. Error bars indicate s.d. (n = 3). *P < 0.05 and **P < 0.01 (two-tailed Student's t-test). **i** The level of PIF4-Myc under the control of *PIF4* promoter is increased by high temperatures in the epidermis. Epidermal tissues were isolated from *PIF4p::PIF4-Myc;pifq* seedlings grown at 20 °C or exposed to 28 °C for 24 h. PIF4-Myc protein in the isolated epidermis was determined by immunoblotting using an anti-Myc antibody. Ponceau S staining is shown for equal loading. The experiments (**a**, **b**, and **i**) were repeated at least two times with similar results. Source data are provided as a Source Data file.

PIF4 to the promoters of these genes was significantly increased by a temperature shift from 16 °C to 28 °C (Fig. 6c). Given that the levels of epidermal PIF4 were not significantly different between the two conditions (Supplementary Fig. 11), the ChIP results indicate that the DNA-binding ability of epidermal PIF4 is enhanced by high temperatures.

**Epidermal *PIF4* expression is induced by high temperatures.** Since *PIF4* mRNA is transcriptionally induced at high temperatures[3,5], it is probable that epidermal *PIF4* expression is also increased by high temperatures. To test this hypothesis, we first compared the tissue-specific GUS signals in *PIF4p::GUS* plants grown either at 16 °C for 5 days or grown at 16 °C and transferred to 28 °C for 24 h. Consistent with previous studies, the GUS signal in *PIF4p::GUS* plants showed a significant overall increase after a

24-h exposure to high temperatures (Fig. 6d). A cotyledon cross-section showed that *PIF4p::GUS* expression was limited mainly to the vasculature and was weak in the epidermis and mesophyll of the plants grown at 16 °C, but *PIF4p::GUS* expression showed an increase both in the epidermis and mesophyll with high-temperature exposure (Fig. 6e). To further confirm that epidermal *PIF4* expression is increased at high temperatures, we isolated the epidermal tissues from the cotyledons of seedlings grown either at 20 °C or transferred to 28 °C for 24 h. The *PIF4* mRNA levels in the epidermal tissues showed an approximate five-fold increase with high-temperature exposure (Fig. 6f). In addition, the PIF4 target genes *YUC8* and *IAA29* were also activated by high temperatures in the epidermis (Fig. 6g, h). We also determined the level of epidermal PIF4 protein regulated by its own promoter (*PIF4p::PIF4-Myc*) in the isolated epidermis. In accordance with the induction of epidermal *PIF4* mRNA, the

level of the epidermal PIF4 protein regulated by the *PIF4* promoter was also increased in response to high-temperature exposure (Fig. 6i). These results indicate that epidermal PIF4 protein levels are increased at high temperatures through the transcriptional activation of *PIF4* expression.

## Discussion

Plant organs are composed of different tissues that are functionally specialized. Therefore, to understand how plants respond to the environmental changes, it is essential to identify the specific functions of each tissue in each environmental response. PIF4 acts as a central signaling hub that integrates multiple environmental signals for growth regulation, and it is expressed in all aerial tissues[15]. Although vascular PIF4 has previously been shown to regulate flowering[38,39], the function of epidermal and mesophyll PIF4 has not yet been determined. In this study, we show that epidermal, but not vascular, PIF4, mediates thermoresponsive hypocotyl growth by characterizing transgenic plants expressing epidermal-specific PIF4 or a dominant negative form of PIF4 (PIF4-SRDX). We also show that high temperatures transcriptionally activate epidermal *PIF4* and enhance epidermal PIF4 DNA-binding ability, which results in the activation of auxin-related genes in the epidermis. Taken together, our findings demonstrate that epidermal PIF4 levels and activity are increased at high temperatures, which then activates epidermal auxin responses in a cell-autonomous manner, thereby facilitating thermomorphogenesis in *Arabidopsis*.

Several studies have demonstrated that the epidermis drives and restricts shoot growth and morphology in plants. Growth promoting hormones such as BR and auxin have been shown to act in the epidermis to determine hypocotyl growth[55,58]. Epidermal phyB was recently shown to be necessary for the light regulation of hypocotyl growth[40]. BR, auxin, and light regulate hypocotyl growth through the modulation of PIF4 activity[15]. Therefore, our finding that epidermal PIF4 mediates thermoresponsive hypocotyl growth is in accordance with the epidermal growth control model. Gibberellic acids (GA) also regulate hypocotyl growth, but the tissue-specific actions of GA on the regulation of hypocotyl growth have not been defined. Since the interaction between DELLAs and PIF4 mediates the GA regulation of hypocotyl growth[34], it is plausible that GA also acts in the epidermis to control hypocotyl growth.

It was previously shown that PIF4 promotes hypocotyl growth partly through the auxin pathway[6,7]. In agreement with this, our transcriptome analyses show that auxin-responsive genes are highly enriched in epidermal PIF4-activated genes (Fig. 2). qRT-PCR analyses with *ML1p::PIF4;pifq* and *ML1p::PIF4ΔN-SRDX* confirm that epidermal PIF4 activity is essential for the high-temperature activation of auxin biosynthetic and responsive genes (Fig. 3). In addition, the block of the epidermal auxin responses suppressed high temperature-induced hypocotyl growth[55] (Fig. 4a). Thus, high temperature appears to activate epidermal auxin biosynthesis and response to trigger thermomorphogenesis including hypocotyl elongation. The high-temperature activation of hypocotyl growth and gene expression was impaired by the overexpression of epidermal phyB (Fig. 4b, c), which could inhibit this process at least in part by directly inhibiting the action of epidermal PIF4. Taken together, these results demonstrate that the epidermal phyB-PIF4-auxin signaling module determines thermomorphogenic growth.

The epidermal auxin response was recently shown to be necessary for the hypocotyl growth response to shade as well as high temperatures[55]. In the epidermis, the auxin response was suggested to promote hypocotyl growth partly by activating the brassinosteroid signaling pathway[55]. However, how shade

activates the epidermal auxin and BR pathways remains poorly understood. Shade-induced hypocotyl growth is also mediated by many PIF factors including PIF4, PIF5, and PIF7[20,59]. These PIFs promote hypocotyl growth through auxin and BR pathways. Epidermal phyB was previously shown to be required for the hypocotyl response to shade[40]. Therefore, it is likely that shade enhances epidermal PIF activities through the inactivation of epidermal phyB, thereby activating epidermal auxin and BR signaling. Given that phyB directly interacts with PIFs and the interaction is required for the inhibition of PIFs[20], epidermal phyB may regulate the activities of epidermal PIF factors probably in a cell-autonomous manner. Our results show that epidermal phyB inhibits thermoresponsive hypocotyl (Fig. 4b). Given that the activity of phyB is regulated by ambient temperatures as well as light, both light and temperature signals are likely to be integrated through epidermal phyB to coordinate plant growth with changes in the environment.

We observed that the epidermal PIF4 protein level was increased by high temperatures when it is regulated by its own promoter, but not by the epidermis-specific *ML1* promoter (Fig. 6). These results suggest that the transcriptional regulation of *PIF4* is a major mechanism underlying high-temperature induction of the epidermal PIF4 protein. In support of this notion, both the *PIF4* promoter-driven GUS signals and the expression of endogenous *PIF4* mRNA showed increases in the epidermis in response to high temperatures. Although the *ML1* promoter-driven PIF4 protein level was not significantly increased, the DNA-binding ability of epidermal PIF4 in *ML1p::PIF4;pifq* plants was significantly increased by high temperature (Fig. 6c), suggesting that high temperatures induce thermomorphogenesis by enhancing epidermal PIF4 DNA-binding as well as epidermal *PIF4* mRNA expression. Consistent with the enhanced epidermal PIF4 binding to its target promoters, the expression of *ATHB2* and *YUC8* was increased by high temperatures in *ML1p::PIF4;pifq* (Fig. 3c, Supplementary Fig. 10). The high-temperature inactivation of phyB may be partly responsible for the enhanced epidermal PIF4 DNA-binding because phyB-PIF4 interaction was previously shown to prevent PIF4 from binding to DNA[56].

Transcriptional regulation of *PIF4* is mediated by the circadian clock. The Evening Complex (EC), consisting of ELF3, ELF4, and LUX, has been reported to mediate the high-temperature activation of *PIF4* expression[12,13]. EC binds directly to the *PIF4* promoter and inhibits its expression, but high temperatures reverse this inhibition by preventing EC from binding to the *PIF4* promoter through unknown mechanisms[11,13,60]. In addition to EC, TOC1 and PRR5 also directly repress *PIF4* expression in the evening[3,61]. Whether these factors also mediate the high-temperature activation of *PIF4* expression remains to be determined. Our results show that epidermal *PIF4* mRNA expression is induced by high temperatures (Fig. 6). It is likely that epidermal ELF3 is directly involved in this process in a cell-autonomous manner. Given that CCA1 directly represses *ELF3* expression[62], the increased *PIF4* expression by epidermal CCA1 overexpression (*CER6p::CCA1*) (Fig. 2d) suggests that epidermal ELF3 could regulate *PIF4* mRNA expression. In support of our hypothesis, epidermal ELF3 driven by the *ML1* promoter significantly repressed hypocotyl growth and partially restored the thermosensitivity of *PIF4* mRNA expression in the *elf3* mutant (Supplementary Fig. 12). Further experiments are required to identify the molecular mechanisms by which epidermal ELF3 activity is regulated by ambient temperatures.

A recent study revealed that PIF7, in addition to PIF4, is required for high temperature-induced hypocotyl growth and gene expression[63]. The PIF7 protein increases in response to elevated ambient temperatures and directly regulates auxin

biosynthetic and signaling genes[63]. Interestingly, PIF7 interacts with PIF4 (ref. [63]). The combination of this interaction and the phenotypes of the *pif4* and the *pif7* single mutants suggests that the PIF4/PIF7 heterodimer may regulate thermomorphogenesis. Epidermal PIF4 is required for thermomorphogenesis (Fig. 3), so there is a high likelihood that PIF7 also acts in the epidermis to control the high-temperature responses through hetero-dimerization with epidermal PIF4.

In conclusion, this study demonstrates the crucial role of epidermal PIF4 in the regulation of thermomorphogenic responses. Collectively, our results suggest that an increase in ambient temperature is perceived in the epidermis probably through phyB and EC, which facilitates the activation of epidermal PIF4. The epidermal PIF4 then induces thermomorphogenesis through the epidermal auxin pathway. A recent study showed that sensing of high temperatures generates a mobile auxin signal in cotyledons, which is required for the promotion of hypocotyl elongation[64]. Our study suggests that the mobile auxin signal is activated by epidermal PIF4 in cotyledons. Future investigations are required to identify the mechanisms underlying the high-temperature regulation of epidermal PIF4 protein level and activity.

## Methods

**Plant materials and growth conditions**. *Arabidopsis thaliana* plants were grown in a greenhouse with a 16 h/8 h light/dark cycle at 22 °C for general growth and seed harvesting. All the *Arabidopsis thaliana* plants used in this study were in a Col-0 ecotype background. To generate transgenic plants with tissue-specific *PIF4* expression, the *ML1* (epidermal) and *SUC2* (vascular) promoters were amplified using the primer pairs indicated in Supplementary Table 1 and fused to the coding sequence of *PIF4*. *ML1p::PIF4* and *SUC2p::PIF4* were cloned into a modified pEarleyGate vector (promoterless version of pEarleyGate 101), and transformed into quadruple *pifq* mutants. To generate the tissue-specific *PIF4ΔN-SRDX* transgenic plants, the *ML1* and *SUC2* promoters were fused to a fragment of *PIF4* lacking its N terminus (amino acid 1–96) and fused to SRDX. *ML1p::PIF4ΔN-SRDX* and *SUC2p::PIF4ΔN-SRDX* were cloned into the modified pEarleyGate vector and transformed into Col-0. To generate *ML1p::PIF4mi* transgenic plants, we used Web MicroRNA Designer 3[52] to design an artificial microRNA sequence (5'-TAAATCGAGGTAACTGTGCCG-3') targeting *PIF4* and transformed the *ML1* promoter-driven artificial microRNA into Col-0. *CER6p::CCA1*, *SUC2p::CCA1*, and *CAB3p::CCA1* mutant seeds were kindly provided by Motomu Endo[46], as were *PIF4p::GUS* seeds[65].

**Hypocotyl length measurement**. Seeds sterilized with 70% (v/v) ethanol and 0.01% (v/v) Triton X-100 were plated on MS medium (PhytoTechnology Laboratories) supplemented with 0.7% phytoagar. After three days of incubation at 4 °C, seedlings were irradiated with white light for 6 h to promote germination and then incubated in specific light (light intensity: 40 μmol m$^{-2}$ s$^{-1}$) and temperature conditions. Seedlings were photocopied and hypocotyl lengths were measured using ImageJ software (http://rsb.info.nih.gov/ij).

**Measurement of hypocotyl negative gravitropism**. To measure hypocotyl negative gravitropism, seeds imbibed for 3 days at 4 °C were grown vertically in the dark for 5 days. Hypocotyl negative gravitropism was measured by calculating the angle of hypocotyl deviation from the vertical axis.

**GUS staining**. For histochemical staining of *PIF4p::GUS*, *PIF4p::GUS* transgenic plants were grown constitutively at 16 °C or at 16 °C and then transferred to 28 °C for 24 h, then immediately treated with 90% acetone and incubated for 30 min at 20 °C. The seedlings were then vacuum-infiltrated in GUS-staining buffer (0.5 mM NaPO4, 10 mM EDTA, 0.5 mM potassium ferrocyanide, 0.5 mM potassium ferricyanide, 1 mM X-Gluc, and 0.1% Triton X-100) and incubated at 37 °C in the dark for 2 to 4 h. The stained tissues were washed and dehydrated using an ethanol series. For cross-sections of cotyledons, GUS-stained seedlings were fixed and then dehydrated using an ethanol series. The cotyledons from the fixed seedlings were embedded in Spurr's resin (Ted Pella) and sectioned using a microtome (MT990).

**Confocal imaging**. Confocal microscopy was performed using Zeiss LSM 710 confocal microscope with 40× water-immersion objective lens. For imaging, seedlings were grown under continuous white light (11w/m², 23 °C) for 4 days on 1/2 Murashige-Skoog medium. After staining the cell walls and nuclei with propidium iodide (PI, diluted to a final concentration of 30 μM with distilled water) for 10 min, seedlings were mounted on a slide glass with distilled water. The emission range of GFP fluorescence was 500 to 540 nm whereas the PI emission range was restricted to 630 to 660 nm in order to minimize the signal overlap between two fluorescence dyes.

**Isolation of epidermal cells**. Epidermal cells were isolated from the seedlings grown in the specific growth conditions as described[66]. Briefly, *Arabidopsis* cotyledons were cut with microscissors and dropped in the 1.5-ml tubes containing 1 ml of enzyme solution (0.75% (wt/vol) cellulase 'onozuka' R-10, 0.25% (wt/vol) macerozyme R-10, 0.4 M mannitol, 8 mM CaCl2 and 5 mM MES-KOH). The tubes were sonicated gently until the cotyledons become transparent and dissolved. The dissolved cotyledons were spread on a petri dish. The epidermis was carefully separated from the vasculature and collected using two syringes with needles under a stereomicroscope. Isolated epidermal tissues were frozen in liquid nitrogen and used for total RNA or protein extraction. The isolation processes were carried out at the same temperature as the growth temperature of the sample plants.

**qRT-PCR gene expression analysis**. Total RNA was extracted from seedlings using the TaKaRa MiniBEST Plant RNA Extraction Kit (Takara) according to the manufacturer's instructions. M-MLV reverse transcriptase (Thermo Fisher Scientific) was used for cDNA synthesis. Quantitative real-time PCR (qRT-PCR) was performed using CFX96 (Bio-Rad) and the EvaGreen master mix (SolGent). Gene expression levels were normalized to that of *PP2A* and are shown relative to the wild-type expression levels. Gene-specific primers are listed in Supplementary Table 1.

**Chromatin immunoprecipitation (ChIP) assays**. Wild type and *ML1p::PIF4;pifq* seedlings kept at 16 °C or shifted to 28 °C for 24 h were cross-linked in 1% formaldehyde solution for 20 min under vacuum. After cross-linking, seedlings were ground in liquid nitrogen. The cross-linked chromatin complex was resuspended by nuclear lysis buffer (50 mM HEPES at pH 7.5, 150 mM NaCl, 1 mM EDTA, 1% Triton X-100, 0.1% Na deoxycholate and 0.1% SDS) and sheared by sonication to reduce the average DNA fragment size to around 0.3 to 0.5 kb. The fragmented chromatin complex was then immunoprecipitated by anti-GFP antibody (GFP-Tag (A-11122) rabbit polyclonal antibody; Thermo Fisher Scientific). The beads were washed with a low-salt buffer (50 mM Tris-HCl at pH 8.0, 2 mM EDTA, 150 mM NaCl and 0.5% Triton X-100), high-salt buffer (50 mM Tris-HCl at pH 8.0, 2 mM EDTA, 500 mM NaCl and 0.5% Triton X-100), LiCl buffer (10 mM Tris-HCl at pH 8.0, 1 mM EDTA, 0.25 M LiCl, 0.5% NP-40 and 0.5% deoxycholate) and TE buffer (10 mM Tris-HCl at pH 8.0 and 1 mM EDTA), and eluted with an elution buffer (1% SDS and 0.1 M NaHCO3). After reverse cross-linking, the PIF4-YFP-bound DNA fragments were purified using a PCR purification kit (Macherey-Nagel). Co-immunoprecipitated DNA fragments were quantified by real-time PCR using specific primers (Supplementary Table 1).

**RNA-Seq analysis**. Seedlings were grown under white light for 5 days at 20 °C. Total RNA was extracted from the seedlings using the MiniBEST Plant RNA Extraction Kit (Takara) according to the manufacturer's instructions. 1 μg of total RNA was processed for preparing mRNA sequencing library using MGIEasy RNA Directional Library Prep Kit (MGI) according to the manufacturer's instruction. Sequencing was conducted on the MGIseq system (MGI) with 100 bp paired-end reads. The quantification of transcript expression was performed using Salmon[67]. Differential expression analysis was performed using DESeq2 software[68]. Differentially expressed genes between samples were defined based on the criteria (expression fold change > 1.5 and adjusted *P*-value < 0.05). Gene ontology enrichment analysis was performed using the Plant GeneSet Enrichment Analysis Toolkit (PlantGSEA)[69].

**Reporting summary**. Further information on research design is available in the Nature Research Reporting Summary linked to this article.

## Data availability

RNA sequencing data were deposited into the Gene Expression Omnibus database under accession number GSE133564 and the NCBI Sequence Read Archive under accession number SRP212662. *Arabidopsis* mutants and transgenic lines used in this study are available upon request from the corresponding author. The source data for Figs. 2a-d, 3a-d, 3f-h, 4a-d, 5a, 6a-c and 6f-i and Supplementary Figs. 2 and 9-11 are provided as a Source Data file.

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

## Acknowledgements

We thank Motomu Endo for providing *CER6p::CCA1*, *SUC2p::CCA1*, *CAB3p::CCA1* and *PIF4p::GUS* seeds and sharing the protocol for the isolation of epidermal cells. This work was supported by grants from the National Research Foundation of Korea (NRF) grant funded by the Korean government (MSIP; Number 2019R1A2C1003783 and 2018R1A3B1052617), the Basic Research Laboratory program of the National Research Foundation funded by the Korean government (grant no. 2017R1A4A1015620), the Next-Generation BioGreen 21 Program (PJ01314801), RDA, Republic of Korea, and a Korea University Grant.

## Author contributions

Sa.K., G.H., So.K., E.O., and G.C. conceived the study and designed the experiments. Sa.K., G.H., T.N.T., H.K., J.J., Ja.K., and So.K. carried out the experiments. E.O., G.C., and Ju.K. supervised the work. Sa.K., G.H., S.K., T.N.T., H.K., G.C., and E.O. wrote the manuscript.

## Competing interests

The authors declare no competing interests.
