## [Peer Review File · Nature Communications]

Reviewers' comments:

Reviewer #1 (Remarks to the Author):

Review of Kim et al.

This manuscript examines the role of PIF4 in different plant tissues in regulating response to temperature. The experiments are well conducted and logical, conclusions are well justified, and the writing is clear. Plants make strong morphological and developmental responses to elevated temperature, including increased elongation and accelerated flowering. Previous work as shown that the transcription factor PIF4 as well as the plant hormone auxin are important for the elongation response and implicated PIF4 in the flowering response. As PIF4 is expressed throughout the plant, it is heretofore unknown which tissues PIF4 acts in to elicit these temperature responses. The authors use multiple lines of evidence to show that epidermal PIF4 is necessary and sufficient for the elongation response. The flowering time response is not explored as extensively but it is clear from the work here that vascular PIF4 plays an important role in regulating flowering.

A few details need attention:

Lines 136 -138. In the absence of mutant data, the statement that the experiments in Fig 2 suggest that increased hypocotyl elongation in CER6p::CCA1 is attributable to increased PIF4 expression is too strong. CCA1 regulates many, many genes; it might be PIF4 or it might be something else (other PIFs?). Or it might be partially PIF4. Reword to say "might be attributable".

Lines 151 - 155. I am not sure that is correct to use the p-values from the hypergeometric test to make arguments about whether vascular or epidermal PIF is more "dominant" in controlling the PIF Quartet global genes. The p-value levels are related not only to the degree of enrichment but also to the size of the two gene categories. IF we examine on a percent basis, 22% (70/309) of the ML1::PIF4 down regulated genes are in the global down set, but only 14% (177/1273) of the SUC2::PIF4 down regulated genes overlap, leading to the opposite conclusion from what is stated in the paper.

Line 186-187 "three times longer". I think this should be "five".

Throughout. "Consistently" is misused, the authors intend "consistent with this fact". There are other minor grammar issues that I assume can be corrected by Nature Communication editorial staff.

Fig 2b: how was expression measured?

Fig 3: please keep the order of genetic lines and constructs the same in all figures and subfigures. The order is swapped in 3f,g,h relative to the rest of the paper.

Line 287: add inhibiting so that the phrase reads "PIF4 is involved in inhibiting the cotyledon opening".

Nozue, Plant Phys, 2011, was the first to present evidence for PIF4 regulating auxin signaling.

Nozue, Nature, 2007, showed that PIF4 is important for diurnal (and circadian) regulation of hypocotyl growth. This is relevant for the CER6p::CCA1 result.

Fig 4b, c: why are ML1::phyB lines less responsive than WT? Is phyB expressed much more strongly from the ML1 promoter than from its endogenous promoter? Please discuss

Fig 6b: is it possible that ML1::PIF4 is expressed so strongly that protein regulation mechanisms are saturated? If not, what is phyB's role in regulating PIF4 during thermogenesis? Solely transcriptional?

Signed: Julin Maloof

Reviewer #2 (Remarks to the Author):

This is a well-written manuscript addressing the roles of epidermal PIF4 in the regulation of hypocotyl elongation responses to ambient temperature. Using a combination of approaches and transgenic lines, the authors show that epidermal PIF4 regulates elongation responses to temperature via the auxin epidermal pathway.

The experimental approach is technically sound, and the paper is well organized. The results are generally clear, and the ideas are logically presented.

My major reservation about this manuscript is that the results appear to be incremental and, in my opinion, they do not represent a step change in our understanding of the mechanisms that regulate hypocotyl growth responses to changes in temperature. In fact, it has been known for a long time that the epidermis plays a key role limiting stem elongation (e.g. Kutschera and Niklas 2007). The authors have previously shown that epidermal phyB represses hypocotyl growth (Kim 2016 TPC). It is also well established that, in Arabidopsis, SAS-like responses to high temperatures require PIF4, and that PIF4 acts by regulating auxin biosynthetic genes (i.e. Franklin's work). In addition, as acknowledged by the authors, work in Chory's lab has demonstrated an important role for the epidermis in auxin-mediated growth of the hypocotyl (i.e. Procko 2016 G&D). Therefore, the authors' demonstration that the epidermis regulates hypocotyl growth responses to temperature through the phyB-PIF4-auxin pathway would appear quite predictable, just as a logical extension of these previous findings.

Other points

The authors use inappropriate statistical analyses to compare the responses to temperature among genotypes. They use student's t tests for each genotype, when the experiments have a clear factorial design. In order to demonstrate that two genotypes have different responses to a given environmental factor (e.g. an increase in temperature), they need to show a significant genotype x temperature interaction. They have not done that for any of their comparisons. In fact, if the data reported by the authors were correctly analyzed, it would be hard to attribute the observed responses to temperature to phyB, as the phyB mutant shows apparently normal responses to temperature in terms of elongation and gene expression (Fig. 4b,c) (from the figures presented by the authors, it appears highly unlikely that the interaction terms are significant in those experiments).

The authors indicate (l. 381) that the high temperature activation of hypocotyl growth and gene expression was impaired by epidermal overexpression of phyB, which is consistent with the roles of phyB as a thermosensor. But epidermal overexpression of phyB would inhibit hypocotyl growth by the local inhibition of PIF4 action, even if phyB was not the temperature sensor. By the way, do we have any idea of the level of overexpression of phyB in the ML1p::PHYB line?

Reviewers' comments:

Reviewer #1 (Remarks to the Author):

Review of Kim et al.

This manuscript examines the role of PIF4 in different plant tissues in regulating response to temperature. The experiments are well conducted and logical, conclusions are well justified, and the writing is clear. Plants make strong morphological and developmental responses to elevated temperature, including increased elongation and accelerated flowering. Previous work has shown that the transcription factor PIF4 as well as the plant hormone auxin are important for the elongation response and implicated PIF4 in the flowering response. As PIF4 is expressed throughout the plant, it is heretofore unknown which tissues PIF4 acts in to elicit these temperature responses. The authors use multiple lines of evidence to show that epidermal PIF4 is necessary and sufficient for the elongation response. The flowering time response is not explored as extensively but it is clear from the work here that vascular PIF4 plays an important role in regulating flowering.

We greatly appreciate the reviewer's comments and suggestions to help improve the manuscript. As you read the revised manuscript with the changes highlighted in yellow, you will see that we tried our best to revise the manuscript by taking into account the comments from the reviewers.

A few details need attention:

Lines 136 -138. In the absence of mutant data, the statement that the experiments in Fig 2 suggest that increased hypocotyl elongation in *CER6p::CCA1* is attributable to increased PIF4 expression is too strong. CCA1 regulates many, many genes; it might be PIF4 or it might be something else (other PIFs?). Or it might be partially PIF4. Reword to say “might be attributable”.

Response: We agree with the reviewer and changed the sentence to

“the hypocotyl elongation seen in *CER6p::CCA1* plants might be attributable to the increased *PIF4* levels in the epidermis”

We also examined the effect of artificial microRNA against *PIF4* (*PIF4mi*) on the hypocotyl growth of *CER6p::CCA1* plants, and we added the results to Supplementary Fig. 2. Down-regulation of *PIF4* mRNA expression by *PIF4mi* largely suppressed the long hypocotyl phenotypes of *CER6p::CCA1* plants. These new results provide support that the increased *PIF4* expression is at least partially responsible for increased hypocotyl elongation in *CER6p::CCA1*.

Lines 151 – 155. I am not sure that is correct to use the p-values from the hypergeometric test to make arguments about whether vascular or epidermal PIF is more “dominant” in controlling the PIF Quartet global genes. The p-value levels are related not only to the degree of enrichment but also to the size of the two gene categories. If we examine on a percent basis, 22% (70/309) of the ML1::PIF4 down regulated genes are in the global down set, but only 14% (177/1273) of the SUC2::PIF4 down regulated genes overlap, leading to the opposite conclusion from what is stated in the paper.

Response: We thank the reviewer for raising this issue. We changed our analysis method from a

hypergeometric test to GSEA (Gene Set Enrichment Analysis), because GSEA results show a more obvious pattern of how the global PIF quartet-activated/repressed genes are regulated by each tissue-specific PIF4. The GSEA plots (Supplementary Fig. 3) support our conclusions.

Line 186-187 “three times longer”. I think this should be “five”.

Response: We have corrected this in the revised manuscript following the suggestion of the reviewer.

Throughout. “Consistently” is misused, the authors intend “consistent with this fact”. There are other minor grammar issues that I assume can be corrected by Nature Communication editorial staff.

Response: We have changed "consistently" to "consistent with this fact," and we have tried to correct minor grammatical errors.

Fig 2b: how was expression measured?

Response: We have measured *PIF4* mRNA expression by performing quantitative real-time PCR with the primers listed below. Data were obtained from three biological replicates, each with two technical replicates. The relative expression levels of *PIF4* were normalized to those of *PP2A*. We have added the experimental details to the corresponding figure legend.

PIF4 F: GCCAAAACCCGGTACAAAACCA
PIF4 R: CGCCGGTGAACATAAATCTCAACATC
PP2A F: TATCGGATGACGATTCTTCGTGCAG
PP2A R: GCTTGGTCGACTATCGGAATGAGAG

Fig 3: please keep the order of genetic lines and constructs the same in all figures and subfigures. The order is swapped in 3f,g,h relative to the rest of the paper.

Response: According to the reviewer’s suggestion, we have made the proposed changes to keep the order of genetic lines the same in all the figures (Fig. 3f, g, h) in the revised manuscript.

Line 287: add inhibiting so that the phrase reads “PIF4 is involved in inhibiting the cotyledon opening”.

Response: We thank the reviewer for the suggestion and have changed the sentence to "epidermal PIF4 is involved in inhibiting the cotyledon opening of etiolated seedlings".

Nozue, Plant Phys, 2011, was the first to present evidence for PIF4 regulating auxin signaling.

Response: Following this suggestion, we have added this in the reference (Line 61).

Nozue, Nature, 2007, showed that PIF4 is important for diurnal (and circadian) regulation of hypocotyl growth. This is relevant for the CER6p::CCA1 result.

Response: We apologize for not citing the paper. We have added the following sentence and the reference (Lines 131-132).

“A previous study showed that *PIF4* expression is constitutively high in CCA1-overexpressing plants (Nozue et al, 2007)”

Fig 4b, c: why are *ML1::phyB* lines less responsive than WT? Is phyB expressed much more strongly from the *ML1* promoter than from its endogenous promoter? Please discuss

Response: The hypocotyls of *ML1p::PHYB* lines are less responsive to high temperatures than those of the wild type probably because overexpression of epidermal phyB by the *ML1* promoter strongly inhibited epidermal PIF4 activity even at high temperatures. To show epidermal phyB is overexpressed in the *ML1p::PHYB* plants, we measured total *PHYB* levels in the wild type and *ML1p::PHYB*, and added this result to Supplementary Fig. 7. Total *PHYB* levels were significantly higher in *ML1p::PHYB* than in the wild type. Given the fact that *PHYB* is ubiquitously expressed in the wild type, but it is expressed only in the epidermis in *ML1p::PHYB* (Kim et al, 2016), this result provides support that epidermal phyB is overexpressed in *ML1p::PHYB* plants compared to the wild type.

We have added the following sentence:

“Total *PHYB* levels were higher in *ML1p::PHYB;phyB-9* than in the wild type (Supplementary Fig. 7), implicating that epidermal phyB is overexpressed in *ML1p::PHYB;phyB-9*”

Fig 6b: is it possible that *ML1::PIF4* is expressed so strongly that protein regulation mechanisms are saturated? If not, what is phyB's role in regulating PIF4 during thermogenesis? Solely transcriptional?

Response: As the reviewer pointed out, epidermal PIF4 may be expressed so strongly in the *ML1::PIF4* plants that *ML1::PIF4* protein stability is not affected much by high temperature. To test this possibility, we examined whether the levels of epidermal PIF4 could be increased by the inactivation of phyB. We determined the levels of epidermal PIF4 in *ML1p::PIF4* seedlings grown under continuous light or shifted to dark. As shown in Supplementary Fig. 9, the epidermal PIF4 protein levels were higher in the dark than under white light, which indicates that the dark-mediated inactivation of phyB could increase the stability of epidermal PIF4. Therefore, it seems that the protein regulation mechanisms are not saturated in *ML1p::PIF4*. In addition, the extent to which epidermal PIF4 is increased by dark was not significantly affected at high temperatures, supporting that epidermal PIF4 protein is not stabilized by high temperatures.

Although the level of *ML1::PIF4* was not altered, the expression of PIF4 target genes (*ATHB2* and *YUC8*) was significantly increased by high temperatures in the *ML1p::PIF4* plants, which suggests that the DNA-binding and/or the transcription activation activity of epidermal PIF4 might be enhanced by high temperatures. To test this possibility, we performed chromatin immunoprecipitation

(ChIP) assays and added the results to Fig. 6c in the revised manuscript. The ChIP results show that epidermal PIF4 binding to its promoters is enhanced by high temperatures. The high-temperature inactivation of phyB could mediate increased PIF4 DNA-binding because phyB-PIF4 interaction was previously shown to prevent PIF4 from binding to its target DNA (Park et al, 2018).

We have added the following sentences in the Discussion:

“Although the *MLI* promoter-driven PIF4 protein level was not significantly increased, the DNA-binding ability of epidermal PIF4 in *MLIp::PIF4;pifq* plants was significantly increased by high temperature (Fig. 6c), suggesting that high temperatures induce thermomorphogenesis by enhancing epidermal PIF4 DNA-binding as well as epidermal *PIF4* mRNA expression. Consistent with the enhanced epidermal PIF4 binding to its target promoters, the expression of *ATHB2* and *YUC8* was increased by high temperatures in *MLIp::PIF4;pifq* (Fig. 4c, Supplementary Fig. 10). The high-temperature inactivation of phyB may be partly responsible for the enhanced epidermal PIF4 DNA-binding because phyB-PIF4 interaction was previously shown to prevent PIF4 from binding to DNA”

Signed: Julin Maloof

Reviewer #2 (Remarks to the Author):

This is a well-written manuscript addressing the roles of epidermal PIF4 in the regulation of hypocotyl elongation responses to ambient temperature. Using a combination of approaches and transgenic lines, the authors show that epidermal PIF4 regulates elongation responses to temperature via de auxin epidermal pathway.

The experimental approach is technically sound, and the paper is well organized. The results are generally clear, and the ideas are logically presented.

We greatly appreciate the reviewer's comments and suggestions to help improve the manuscript. As you read the revised manuscript with the changes highlighted in yellow, you will see that we have tried our best to revise the manuscript by taking into account the comments from the reviewers.

My major reservation about this manuscript is that the results appear to be incremental and, in my opinion, they do not represent a step change in our understanding of the mechanisms that regulate hypocotyl growth responses to changes in temperature. In fact, it has been known for a long time that the epidermis plays a key role limiting stem elongation (e.g. Kutschera and Niklas 2007). The authors have previously shown that epidermal phyB represses hypocotyl growth (Kim 2016 TPC). It is also well established that, in Arabidopsis, SAS-like responses to high temperatures require PIF4, and that PIF4 acts by regulating auxin biosynthetic genes (i.e. Franklin's work). In addition, as acknowledged by the authors, work in Chory's lab has demonstrated an important role for the epidermis in auxin-mediated growth of the hypocotyl (i.e. Procko 2016 G&D). Therefore, the authors' demonstration that the epidermis regulates hypocotyl growth responses to temperature through the phyB-PIF4-auxin pathway would appear quite predictable, just as a logical extension of these previous findings.

Response:

As the reviewer pointed out, it was previously shown that epidermal phyB represses hypocotyl growth. However, the molecular mechanism of how epidermal phyB regulates hypocotyl growth is still unknown. Although many components have been identified to regulate hypocotyl growth in the phyB signaling pathway, whether these components act in the epidermis has not been examined. Besides, given that epidermal phyB is able to regulate endodermal PIF1 (a PIF4 homolog) through an unidentified mobile signal (Kim et al, 2016), there is a possibility that epidermal phyB could control the downstream components in a non-cell autonomous manner to regulate hypocotyl growth.

Chory's lab has shown that the epidermis is important for auxin-mediated hypocotyl growth. However, how the epidermal auxin biosynthesis and signaling are regulated by environmental changes (e.g., high temperatures) is not well understood. Although it is likely that the epidermis directly senses an elevation in ambient temperatures and activates epidermal auxin signaling in a cell-autonomous manner, it is also possible that auxin synthesized in other tissues is transported to the epidermis at high temperatures since auxin is a mobile hormone.

In this manuscript, we found that PIF4 acts in the epidermis to regulate thermoresponsive hypocotyl growth. In addition, we demonstrated that epidermal PIF4 mediates the epidermal phyB regulation of hypocotyl growth, and high temperatures activate epidermal auxin responses through epidermal PIF4. Therefore, our manuscript elucidates that the signaling module consisting of epidermal phyB-PIF4-auxin controls thermoresponsive hypocotyl growth. Such a growth regulation mechanism by epidermal PIF4 has not been suggested before, and it has not been experimentally demonstrated either. Moreover, considering it is one of many possible mechanisms as we explained above, our findings cannot be regarded as quite predictable.

In the revised manuscript, we further show that epidermal PIF4 DNA-binding is enhanced by high temperatures (Fig. 6c), and epidermal ELF3 partially restores the thermoresponsiveness of *PIF4* mRNA expression in the *elf3* mutant (Supplementary Fig. 12). Given that ELF3 DNA-binding is reduced at high temperatures (Ezer, D. et al, (2017) *Nat Plants*), these results, together with the fact that the epidermal *PIF4* expression increases at high temperatures, suggests that the epidermis directly senses the changes in ambient temperatures and determines growth responses through the epidermal phyB-PIF4-auxin pathway. As a multicellular organism, each tissue (or organ) in plants has distinct functions, and thus differentially responds to the same environmental change. Therefore, the elucidation of tissue-specific functions/responses is essential to understand how plants respond to environmental changes correctly. However, the tissue-specific functions in the regulation of growth response to temperature changes and underlying molecular mechanisms have not been addressed yet. In this regard, we believe our manuscript addresses an important question and significantly advances our understanding of the response of plants to temperature changes.

Other points

The authors use inappropriate statistical analyses to compare the responses to temperature among genotypes. They use student's t tests for each genotype, when the experiments have a clear factorial design. In order to demonstrate that two genotypes have different responses to a given environmental factor (e.g. an increase in temperature), they need to show a significant genotype x temperature interaction. They have not done that for any of their comparisons. In fact, if the data reported by the authors were correctly analyzed, it would be hard to attribute the observed responses to temperature to phyB, as the phyB mutant shows apparently normal responses to temperature in terms of elongation and gene expression (Fig. 4b,c) (from the figures presented by the authors, it appears highly unlikely that the interaction terms are significant in those experiments).

Response: We thank the reviewer for the suggestion and have reanalyzed the results (Fig. 3a, d, f and Fig. 4b, c, d) with a two-way ANOVA to show that two genotypes have differential responses to high temperature. New analyses indicate that the high-temperature response of hypocotyl growth, but not *YUC8* expression, is significantly different between WT and *phyB* (Fig. 4b, c). In contrast, both responses were significantly different between WT and *ML1p::PHYB*, indicating that the overexpression of epidermal *phyB* indeed affects (suppresses) thermomorphogenesis.

The authors indicate (l. 381) that the high temperature activation of hypocotyl growth and gene expression was impaired by epidermal overexpression of *phyB*, which is consistent with the roles of *phyB* as a thermosensor. But epidermal overexpression of *phyB* would inhibit hypocotyl growth by the local inhibition of PIF4 action, even if *phyB* was not the temperature sensor. By the way, do we have any idea of the level of overexpression of *phyB* in the *ML1p::PHYB* line?

Response: We agree with the reviewer and have changed the sentences as follows:

“The high-temperature activation of hypocotyl growth and gene expression was impaired by the overexpression of epidermal *phyB* (Fig. 4b,c), which could inhibit this process at least in part by directly inhibiting the action of epidermal PIF4”

To show epidermal *phyB* is overexpressed in the *ML1p::PHYB*, we have measured total *PHYB* levels in the wild type and *ML1p::PHYB*, and we have added this result to Supplementary Fig. 7. Total *PHYB* levels were significantly higher in *ML1p::PHYB* than in the wild type. Given that *PHYB* is ubiquitously expressed in the wild type, but it is expressed only in the epidermis in *ML1p::PHYB* (Kim et al, 2016), this result indicates that epidermal *phyB* is highly overexpressed in *ML1p::PHYB* transgenic plants compared to the wild type.

Reviewers' comments:

Reviewer #1 (Remarks to the Author):

This interesting paper shows that PIF4 functions specifically in the epidermis to regulate thermoresponsive growth. The authors have nicely addressed my concerns with the first version. New experiments in this version further elucidate the mechanisms of PIF4 action in the epidermis in response to high temperature.

Reviewer #2 (Remarks to the Author):

The authors have responded to my previous comments. Regarding my major reservation, I can understand the authors' point of view, but I still see the results as incremental, and more appropriate for a more specialized journal. This is, of course, an editorial decision and I will not add further comments on this point.

Regarding my comments on the use of inappropriate statistical analyses, I do not see much progress in the current version. These are all factorial experiments. Therefore, they should be analyzed using a factorial ANOVA, and the authors should report the significance of the "interaction term". If the interaction (e.g. between genotype and temperature) is significant (and only if it is significant), then the analysis demonstrates that the "response" to temperature differs between (or among) genotypes. Only in that case, it is appropriate to run comparisons between means to test whether two particular genotypes are differentially affected by a temperature shift. The current presentation does not show the P-values for the interaction terms of the ANOVAs, and it has a curious mixture of ANOVA tests and t-tests that is not correct. Please, redo the analyses and show the P-values for the interaction terms to support the claims that the different genotypes show different responses to temperature.

In their response to my previous comments, the authors indicate "New analyses indicate that the high-temperature response of hypocotyl growth, but not YUC8 expression, is significantly different between WT and phyB (Fig. 4b, c). In contrast, both responses were significantly different between WT and ML1p::PHYB, indicating that the overexpression of epidermal phyB indeed affects (suppresses) thermomorphogenesis." Yet, the P-values for interaction terms are not shown. The phyB mutant shows an apparently very robust response to temperature. It is admittedly not as high as that of the WT (in relative terms), but it is clearly present. Therefore, it would be important to show that this apparently reduced response is not just a consequence of the fact that phyB hypocotyls can simply not elongate much faster than that. Can the Arabidopsis hypocotyls grow much longer than 8 mm within the time frame of these experiments? This needs to be tested. The data for the ML1p::PHYB lines are equally unhelpful. They simply show that overexpressed epidermal phyB suppresses elongation, which is only to be expected based on Kim et al. 2016 –TPC. Because of these limitations in the analysis and interpretation of results, I do not think that the data presented here support the conclusions expressed in the titles of Figs. 3 and 4: "Epidermal PIF4 activity is required for thermoresponsive hypocotyl growth" and "Epidermal auxin and phyB regulate high temperature-induced hypocotyl growth"

Finally, some mention needs to be made to recent data showing the PIF7 is also involved in the response to elevated temperature in Arabidopsis (see Fiorucci et al. 2019 –New Phyt).

Reviewer #1 (Remarks to the Author):

This interesting paper shows that PIF4 functions specifically in the epidermis to regulate thermoresponsive growth. The authors have nicely addressed my concerns with the first version. New experiments in this version further elucidate the mechanisms of PIF4 action in the epidermis in response to high temperature.

Response: We thank the reviewer for the helpful suggestion, which improved the manuscript substantially.

Reviewer #2 (Remarks to the Author):

The authors have responded to my previous comments. Regarding my major reservation, I can understand the authors' point of view, but I still see the results as incremental, and more appropriate for a more specialized journal. This is, of course, an editorial decision and I will not add further comments on this point.

Regarding my comments on the use of inappropriate statistical analyses, I do not see much progress in the current version. These are all factorial experiments. Therefore, they should be analyzed using a factorial ANOVA, and the authors should report the significance of the "interaction term". If the interaction (e.g. between genotype and temperature) is significant (and only if it is significant), then the analysis demonstrates that the "response" to temperature differs between (or among) genotypes. Only in that case, it is appropriate to run comparisons between means to test whether two particular genotypes are differentially affected by a temperature shift. The current presentation does not show the P-values for the interaction terms of the ANOVAs, and it has a curious mixture of ANOVA tests and t-tests that is not correct. Please, redo the analyses and show the P-values for the interaction terms to support the claims that the different genotypes show different responses to temperature.

Response: We appreciate these helpful suggestions.

In the previous revision, we performed pairwise two-way ANOVAs (e.g., between specific two genotypes; WT and *phyB-9* or WT and *ML1p::PHYB;phyB-9*) to examine whether the temperature response significantly differed between two particular genotypes and provided P-values for the interaction terms of the pairwise two-way ANOVAs at the top of Figs. 3 and 4 (red asterisks). We now realize that this was not clear. As the reviewer suggested, we have now analyzed the data with two-way ANOVAs across all genotypes and temperatures and added the P-values for the interaction terms (Gen*temp) at the top of Figs. 3 and 4. We have also provided the relevant descriptions in the figure legends.

In their response to my previous comments, the authors indicate "New analyses indicate that the high-temperature response of hypocotyl growth, but not YUC8 expression, is significantly different between WT and *phyB* (Fig. 4b, c). In contrast, both responses were significantly different between WT and *ML1p::PHYB*, indicating that the overexpression of epidermal *phyB* indeed affects (suppresses) thermomorphogenesis." Yet, the P-values for interaction terms are not shown.

Response: As the reviewer suggested, we added the P-values for interaction terms (Gen*temp) of the two-way ANOVAs across all genotypes and temperatures at the top of Fig. 4. The analyses revealed significant interactions between genotypes and temperatures (e.g., P-value for interaction term < 0.0001 in Fig. 4b and $P = 0.001$ in Fig. 4c), indicating that different genotypes show different temperature responses.

The *phyB* mutant shows an apparently very robust response to temperature. It is admittedly not as high as that of the WT (in relative terms), but it is clearly present. Therefore, it would be important to show that this apparently reduced response is not just a consequence of the fact that *phyB* hypocotyls can simply not elongate much faster than that. Can the *Arabidopsis* hypocotyls grow much longer than 8 mm within the time frame of these experiments? This needs to be tested.

Response: To address the issue raised by the reviewer, we measured the hypocotyl lengths of the *phyABCDE* quintuple mutant. Our results (Reviewer Only Fig. 1) show that the hypocotyls can grow to about 14 mm in the same condition described in Fig. 4b. This indicates that the reduced response of the *phyB* mutant is not simply because the *Arabidopsis* hypocotyls cannot grow longer than 8 mm. This result is well consistent with the proposed functions of *phyB* in thermomorphogenesis; the inactivation of *phyB* by high temperatures contributes to the increase in hypocotyl growth, so that the difference in hypocotyl lengths between two temperatures is reduced in the *phyB* mutant (Jung et al, 2016; Legris et al, 2016; Qiu et al, 2019). The main finding of the experiments with the *ML1p::PHYB;phyB* lines is that epidermal *phyB* inhibits thermoresponsive growth, not that *phyB* is involved in this process; the latter has been demonstrated (Jung et al, 2016; Legris et al, 2016; Qiu et al, 2019).

Reviewer Only Fig. 1

The data for the *ML1p::PHYB* lines are equally unhelpful. They simply show that overexpressed epidermal *phyB* suppresses elongation, which is only to be expected based on Kim et al. 2016 –TPC. Because of these limitations in the analysis and interpretation of results, I do not think that the data presented here support the conclusions expressed in the titles of Figs. 3 and 4: “Epidermal PIF4 activity is required for thermoresponsive hypocotyl growth” and “Epidermal auxin and *phyB* regulate high temperature-induced hypocotyl growth”

Response: We do not agree that the *ML1p::PHYB* data are unhelpful. Although it was previously shown that *phyB* controls thermomorphogenesis (Jung et al, 2016; Legris et al, 2016; Qiu et al, 2019), the tissue where *phyB* regulates the temperature responses have never been identified. Our results that high temperature-induced hypocotyl growth and the expression of the PIF4 target gene (*YUC8*) were impaired in the *ML1p::PHYB* plants indicate that epidermal *phyB* activity determines the high-temperature responses. Since *phyB* functions as a thermosensor, the results suggest that the epidermis senses temperature changes for growth regulation.

The results also support the hypothesis that epidermal PIF4 activity is required for thermoresponsive hypocotyl growth because *phyB* directly inhibits PIF4 (Part et al, 2018; Lorrain et al, 2008). We provide evidence to further support this idea:

- Epidermal PIF4 (*ML1p::PIF4;pifq*) promotes hypocotyl growth and the expression of auxin/cell growth-related genes.
- The inactivation of epidermal PIF4 (*ML1p::PIF4mi* and *ML1p::PIF4ΔN-SRDX*) suppresses thermoresponsive hypocotyl growth and gene expression.
- Epidermal PIF4 binds to the promoters of the auxin-genes, and its DNA-binding is enhanced by high temperatures.
- The expression level of epidermal PIF4 increases with high temperatures.

Finally, some mention needs to be made to recent data showing the PIF7 is also involved in the response to elevated temperature in Arabidopsis (see Fiorucci et al. 2019 –New Phyt).

Response: As the reviewer suggested, we have added the following sentences to the Discussion.

“A recent study revealed that PIF7, in addition to PIF4, is required for high temperature-induced hypocotyl growth and gene expression. The PIF7 protein increases in response to elevated ambient temperatures and directly regulates auxin biosynthetic and signaling genes. Interestingly, PIF7 interacts with PIF4. The combination of this interaction and the phenotypes of the *pif4* and the *pif7* single mutants suggests that the PIF4/PIF7 heterodimer may regulate thermomorphogenesis. Epidermal PIF4 is required for thermomorphogenesis (Fig. 3), so there is a high likelihood that PIF7 also acts in the epidermis to control the high-temperature responses through hetero-dimerization with epidermal PIF4.”

REVIEWERS' COMMENTS:

Reviewer #2 (Remarks to the Author):

The authors have now applied proper statistical analyses and interaction terms are shown for factorial experiments. This represents an improvement. Fig. 3 is really nice and the whole manuscript is well organized. That said, I highlight that we already knew that: 1) epidermal phyB represses hypocotyl growth in response to light (Kim 2016); 2) growth responses to high temperatures require PIF4; 3) phyB effects on growth are mediated through PIF inactivation; 4) PIF4 acts by promoting auxin biosynthetic genes; 5) hypocotyl elongation responses to shade and temperature depend on epidermal auxin (Procko 2016). Therefore, the idea that hypocotyl growth responses to temperature are controlled by the epidermis through the phyB-PIF4-auxin pathway would appear to be an obvious (albeit not fully tested) hypothesis. In fact, this is to some extent acknowledged by the authors, when they write (l 264-) "It was previously shown that epidermal phyB represses hypocotyl growth (ref to Kim). Given our results indicating that epidermal PIF4 promotes hypocotyl growth, it is highly likely that epidermal phyB impacts on hypocotyl growth by repressing the epidermal PIF4 activity." This "highly likely" hypothesis is tested in Fig. 4 and the results are generally consistent with this idea ---although it must be noted that the phyB-9 mutant still shows a strong and highly significant response to temperature, indicating that phyB is not essential for the temperature response in the author's experimental conditions. The authors show that epidermal PHYB overexpression suppresses the growth of the hypocotyl at high temperature, which is totally consistent with the observation of Kim et al, who demonstrated (using similar or identical lines) that epidermal phyB controls the response to R light (Fig. 3 in Kim et al.). Assuming that at least part of the temperature response is mediated by phyB under the conditions used by the authors (and also assuming that the phenotype of the ML1p::PHYB line is not simply a consequence of very high levels of PHYB overexpression), it is only to be expected that the observations of Kim et al. for light responses would be confirmed when the status of phyB is modified using a temperature shift instead of a light signal. This is because the phyB-mediated effects of light and temperature essentially depend on the same change in the photoreceptor status (changes in Pfr).

Reviewer #2 (Remarks to the Author):

The authors have now applied proper statistical analyses and interaction terms are shown for factorial experiments. This represents an improvement. Fig. 3 is really nice and the whole manuscript is well organized.

Response: We thank the reviewer for the helpful suggestion, which improved the manuscript substantially.

That said, I highlight that we already knew that: 1) epidermal phyB represses hypocotyl growth in response to light (Kim 2016); 2) growth responses to high temperatures require PIF4; 3) phyB effects on growth are mediated through PIF inactivation; 4) PIF4 acts by promoting auxin biosynthetic genes; 5) hypocotyl elongation responses to shade and temperature depend on epidermal auxin (Procko 2016). Therefore, the idea that hypocotyl growth responses to temperature are controlled by the epidermis through the phyB-PIF4-auxin pathway would appear to be an obvious (albeit not fully tested) hypothesis. In fact, this is to some extent acknowledged by the authors, when they write (1264-) “It was previously shown that epidermal phyB represses hypocotyl growth (ref to Kim). Given our results indicating that epidermal PIF4 promotes hypocotyl growth, it is highly likely that epidermal phyB impacts on hypocotyl growth by repressing the epidermal PIF4 activity.” This “highly likely” hypothesis is tested in Fig. 4 and the results are generally consistent with this idea --- although it must be noted that the phyB-9 mutant still shows a strong and highly significant response to temperature, indicating that phyB is not essential for the temperature response in the author’s experimental conditions. The authors show that epidermal PHYB overexpression suppresses the growth of the hypocotyl at high temperature, which is totally consistent with the observation of Kim et al, who demonstrated (using similar or identical lines) that epidermal phyB controls the response to R light (Fig. 3 in Kim et al.). Assuming that at least part of the temperature response is mediated by phyB under the conditions used by the authors (and also assuming that the phenotype of the ML1p::PHYB line is not simply a consequence of very high levels of PHYB overexpression), it is only to be expected that the observations of Kim et al. for light responses would be confirmed when the status of phyB is modified using a temperature shift instead of a light signal. This is because the phyB-mediated effects of light and temperature essentially depend on the same change in the photoreceptor status (changes in Pfr).

Response: Although epidermal phyB was previously shown to mediate light-regulated hypocotyl growth, whether it also mediates temperature-regulated hypocotyl growth has never been experimentally demonstrated. The signaling mechanisms downstream of epidermal phyB are not defined. How the epidermal auxin is regulated by shade and temperature is also not well understood. Our genetic, genomic, and biochemical results illustrate that temperature is sensed by the epidermis (partly by phyB), which stimulates epidermal auxin responses and thermoresponsive hypocotyl growth through direct regulation of epidermal PIF4. This epidermal phyB-PIF4-auxin hypothesis is one of many possible mechanisms, because 1) phyB regulates many components in addition to PIF4, 2) phyB can affect the activity of one of the PIFs in a non-cell-autonomous manner (Kim et al., 2016), and 3) auxin is a well-known mobile hormone. Our study not only provides strong evidence that the epidermis is a critical tissue in the temperature-regulated hypocotyl growth but also elucidates a complete signaling pathway from a temperature receptor to a growth hormone in the epidermis. Therefore, we believe our study significantly advances our understanding of how plants react to temperature changes.